# The economic value of Canada's National Capital Green Network

Chloé L'Ecuyer-Sauvageau[1]*, Jérôme Dupras[1], Jie He[2], Jeoffrey Auclair[1], Charlène Kermagoret[1], Thomas G. Poder[3,4]

1 Département des sciences naturelles, Université du Québec en Outaouais, Ripon, Québec, Canada, 2 Département d'économique, École de Gestion, Université de Sherbrooke, Sherbrooke, Québec, Canada, 3 École de santé publique–Département de gestion, d'évaluation et de politique de santé, Université de Montréal, Montréal, Québec, Canada, 4 Centre de recherche de l'Institut universitaire en santé mentale de Montréal, CIUSSS de l'Est de l'île de Montréal, Montréal, Québec, Canada

☯ These authors contributed equally to this work.
* lecc23@uqo.ca

**Data Availability Statement:** All relevant data are within the paper and its Supporting Information files.

## Abstract

The lack of information on the value of ecosystems contributing to human well-being in urban and peri-urban setting is known to contribute to the degradation of natural capital and ecosystem services (ES). The purpose of this study was to determine the economic value of ES in Canada's Capital Region (Ottawa-Gatineau region), so that these values can be integrated in future planning decisions. Using the valuation methods of market pricing, cost replacement, and two benefit transfer approaches (with adjustment and with meta-analysis), the value of 13 ES from five ecosystems (forests, wetlands, croplands, prairies and grasslands, and freshwater systems) was measured. The annual economic value of these 13 ES amounts to an average of 332 million dollars, and to a total economic value of over 5 billion dollars, annualized over 20 years. The largest part of this value is generated by nonmarket ES, indicating that much more emphasis should be put on the management, preservation, and understanding of processes that make up these types of ES. The work generated as part of this study is a first step towards operationalizing the concept of ES in planning. More specifically, these results can be used to raise awareness, but also as a stepping stone to improve ecosystem-wide planning in the Canada's Capital Region.

## Introduction

In 2014, 54% of the world's population lived in cities, with this percentage expected to reach 66% by 2050 [1], thus making the management of urban areas one of the top development challenges of the 21st century [1]. The management of urban areas involves many elements, ranging from transportation to health, but also including justice and environmental issues. Urban sprawl is a side effect of the attraction of humans to urban centres, but the extent of the sprawl is a function of land-use controls, municipal fragmentation, housing prices, reliance of municipalities on property taxes to finance public services, and the percentage of public spending on infrastructure and transportation [2]. Although there are many definitions of urban sprawl, common elements of this definition in the North American context include low-

**Funding:** The study was funded by the National Capital Commission (NCC) and by the Social Sciences and Humanities Research Council of Canada. The funders had no role in study design, data collection and analysis, decision to publish, or preparation of the manuscript. Jie He thanks the financial support from Guangzhou Natural Science Fund (201904010189). The financial support from the Guangzhou Natural Science Fund did not influence the study design, data collection and analysis, decision to publish, or preparation of the manuscript.

**Competing interests:** The authors have declared that no competing interests exist.

density developments, reliance on the use of private automobiles, a lack of functional open space, waste of land, segregated land uses, and the presence of a homogeneous population [3–7]. The urban sprawl phenomenon is having impacts on air pollution, especially from transportation, increased traffic congestion, and it is characterized by a wasteful use of resources [4, 8, 9].

Influenced by the works of Ebenezer Howard and the Garden City movement [10, 11], the cities of Ottawa and Gatineau have implemented the Gréber Plan in 1950. The main intentions behind this plan was to control urban growth, improve the beauty of Canada's Capital Region, and put aside land for future institutional and agricultural uses, as well as for recreational use [12]. These lands included a Greenbelt around the City of Ottawa, the Gatineau Park on the Quebec side, and many natural areas within the City of Ottawa and along main waterways. Today, this Green Network is managed by the National Capital Commission (NCC), a federal agency of the Canadian government.

One of the ways to address environmental and well-being issues associated with urban sprawl is through the lens of ecosystem services (ES). ES, as defined by the MEA [13], are functions and ecological processes generated by ecosystems that benefit humans in many aspects of their lives. These benefits can be direct, through the provision of basic goods, but also indirect, through their impact on constituents of well-being, i.e. on security, positive social relations, health, and the freedom of choice.

The concept of ES also enables decision makers, public and private, to reflect on the impact of planning decisions on the quality of life of citizens through the use of new indicators. According to the MEA's [13] classification, there are four categories of ES: regulating, supporting, provisioning and cultural. In an urban and peri-urban context, these classes of services can be represented, for example, by trees and forests that have the ability to capture and store carbon, provide shade, act as habitats for biodiversity, generate timber and non-timber forest products, and have an aesthetic value.

The quality and quantity of ES is especially degraded in urban and peri-urban settings as a result of a failure to recognize their importance and to systematically include them in planning [14–16]. This situation is especially true for ES that have no monetary value in the traditional market. One way to remediate this problem is to determine the economic value of ES and of natural capital, to demonstrate their contribution to human well-being and to enable a true benefit-cost analysis of different planning options. Many cities around the world, like New York and Toronto, have integrated the concept of ES in their land-use plans [17, 18].

The purpose of this study was to determine the economic value of ES and natural capital in the Ottawa-Gatineau region (i.e. Green Network of Canada's Capital Region), so that these values can be integrated in future planning decisions. A technical report was published by Dupras et al. [S1 File], and presented preliminary results for the ES valuation of the Canada's National Capital Green Network. In this article, we deepen the methodological and discussion aspects of the study. To do so, there is a Site description section, a Material and methods section, a Results section, a Discussion section which will focus on the way our results could impact future environmental management and planning decisions in the region, and a Conclusion.

## Site description

The Capital of Canada is located in Ottawa, Ontario, but the National Capital Region (NCR) includes cities from both the provinces of Quebec and Ontario. The most important cities of the NCR are Ottawa and Gatineau, which are separated by the Ottawa River, but linked by five inter-provincial bridges and an economy mainly based on federal public services. The NCR also includes 12 cities and municipalities. The population in the metropolitan region of

Ottawa-Gatineau was estimated at 1,329,807 people in 2015 [19]. The region covers an area of 6,767 km$^2$ with a population density of 195.6 people per square kilometre [20]. The GDP per capita in the Ottawa-Gatineau metropolitan region was approximately $48,500 in 2015 [19, 21, 22]. In 2014 the total median income was $87,060 per household on the Quebec side of the region, and $102,030 per household on the Ontario side [23].

The NCR is located in a temperate climate zone, with warm summers and cold and snowy winters. It is also located in the Ottawa River watershed which covers an area of 146,334 km$^2$ [24].

The creation of the Gatineau Park precedes the Gréber Plan in 1950. The park's appeal for citizens started at the end of the 1800s with recreational activities being carried out in the park and rich individuals building cottages in the hills close to the city [25]. It only gained an official statute in 1938, after the beginning of the Great Depression and the emergence of conflicts between recreational users and resource harvesters. It was created to promote recreational activities and to be used as a conservation area [26]. It is still a prime destination for recreational and cultural activities; more than 2.6 million visits are made to the park annually [27]. The park hosts diverse habitats and has a high level of biodiversity. It contains 50 lakes and hundreds of ponds, mixed, deciduous and coniferous forests, there are also approximately 1,600 floral species, 54 mammals, 232 avian, 17 amphibians, 11 reptiles, more than 50 fish species and many species at risk [28].

The 1950 Gréber Plan recommended the creation of a Greenbelt and the development of a network of urban lands that would be connected to the existing Gatineau Park. The main objectives of the Greenbelt were to contain population growth, up to 500,000 citizens, provide natural areas for the enjoyment of the urban population, and act as a reserve of lands for agriculture and for governmental institutions [12]. The first objective was not achieved, especially because of the low capacity of the inner city. The Greenbelt was originally implemented around the city, but in 2001, twelve local administrations within and outside the greenbelt were merged, with the old City of Ottawa representing only the inner portion. Today, because of the number of people living outside the boundaries of the Greenbelt but working within its limits, transportation is one of the most important challenges to the health of the Greenbelt, in addition to population growth [29].

The creation of the Greenbelt was mainly carried out by the NCC through land purchases and expropriations [12]. Today, the Greenbelt has a spatial extent of approximately 20,600 hectares, composed mainly of natural areas, agricultural lands and forests, but also including roads, an airport, and residential and institutional areas. Recreational and cultural activities are carried out in the Greenbelt, which attracts 3.5 million visitors annually [29]. Conservation of natural areas is also important in the Greenbelt, and the most common habitats are wetlands and significant forests. For example, Mer Bleue is recognized as an internationally significant wetland under the Ramsar convention [29]. There are few old-growth forests in the Greenbelt because of agriculture and urbanization, but the existing ones are very important as seed sources for late succession tree species. There are also many species living in the Greenbelt, including 60 species at risk.

Natural Urban Lands make up for a small proportion of the NCC's Green Network, i.e. 4,500 hectares. These lands are used mainly for recreational and access purposes, but they also provide habitats for species in the form of a heron nesting site, fish spawning habitats and islands that are used by birds as refuges in the urban lands. These lands are also home to more than 70 species at risk.

As part of this study, we consider that the natural capital of the Green Network of the Canada's Capital Region is represented by all lands managed by the NCC, which includes the Greenbelt, the Gatineau Park and natural urban lands. The NCC being one of the largest

owners of natural lands in the area, i.e. more than 55,000 hectares, we decided to focus our study on the lands managed by this entity. Common challenges to the health of Ottawa and Gatineau's natural areas include climate change, encroachment on natural environments, habitat fragmentation and invasive species [28].

Having green spaces within and around cities can increase the resilience of our environments. It can be difficult, however, for city planners to reconcile environmental and development pressures. In Canada, municipalities get funding from the provinces to carry out tasks that are delegated to them. To get additional funding, municipalities have the ability to raise taxes, but a large part of their income comes from property taxes, which can only be raised if houses are built. In this context, cities tend to open up land for housing development, without considering the impact to their natural capital. In addition, it is often politically damaging for municipal governments to raise new taxes or to increase the taxation level of citizens. The case of the NCC's Green Network is singular in the sense that the largest green network in the Ottawa-Gatineau region is not managed by the municipalities, but by an agency of the federal government that is not influenced by the same pressures. It can, however, be indirectly influenced by political pressures, through budget allocation and changes to its mandate. In this case, assessing the economic value of the lands preserved by the NCC, can help to showcase the importance of natural capital to human well-being in the region.

## Materials and methods

Many methods exist to determine the economic value of ES, including methods based on market pricing, replacement cost, revealed and stated preferences, as well as benefit transfer.

There is a distinction to be made between the evaluation of stocks of natural resources and of flow of ES. For Jones et al. [30], natural capital encompasses the 'stocks', which represent assets found in the environment, and the processes through which humans perceive benefits, which can be regarded as 'flows' or as transformations or evolutions of the stocks [30: 154]. Whereas the stock value of a wetland would be its replacement cost, the flow of the services provided by wetlands include flood prevention, carbon sequestration and habitat for biodiversity. In this study, we did include the values of the stock of wetlands, as the meta-analysis by He et al. [31] provides the value of the stock of one ha. When it comes to carbon storage, the value of the stock of carbon was taken into account for the service of Global climate regulation for Forests, Wetlands, and Prairies and grasslands. However, the value of the stock of capital was not included in the value of the other ES.

In this study, we used a combination of market pricing method and two benefit transfer methods (i.e. meta-analysis and benefit transfer with adjustment) for the year 2015. The latter are based on the use of secondary data and are frequently used for these types of studies (see Dupras and Alam [9] for a review). This choice of methods was based in part on time and resource constraints, and in part on the purpose of this study, which was to raise awareness on the general importance of ES in the NCC's Green Network.

Our approach follows Troy and Wilson's [32], which starts by choosing the ES to value based on a spatial analysis of the study area (identify land use types). Then, once the goods and ES have been identified by land use types, including whether they are urban, peri-urban or rural, the mode of evaluation of ES is chosen, based on available resources. As a third step, we collected data that enabled us to determine the economic value of ES that had been previously identified. The data collection included research on market prices for provisioning services, spatial data collection on wetlands to carry out the benefit transfer using the meta-analysis method by He et al. [31], a literature review and the creation of a database for services not covered under the market pricing and benefit transfer methods. Using the data collected, we have

determined the value of ES based on the ecosystems present, adapted these values to the land cover, and created maps showing the results.

## Spatial analysis

The land cover analysis was performed using Agriculture and Agri-Food Canada's (AAFC) 2014 land cover inventory database [33]. This layer was the most appropriate geographic information system (GIS) database as it included many land use classes (28 categories) and allowed for a consistent coverage of the entire study area at a fine resolution (30 m$^2$). This satellite imagery has a level of precision of 85%. Although the study area was too large to carry out manual correction, which would have increased the accuracy of the GIS analysis, the use of a randomized photo-interpretation approach confirmed the general preciseness of the layer and its suitability for subsequent analysis. This layer was coupled with Statistics Canada's classification of urban and rural areas, to differentiate between these environments. The criteria used to distinguish urban and rural areas is based on demographic characteristics (population size and density) and distance to large agglomerations. For our purposes, an urban area has a population size of at least 1,000 people and a population density of at least 400 people per square kilometre. The land cover analysis was carried out using the ArcGIS software (Esri, Redlands, CA, United States).

## Selection of ecosystem services

The selection of ES was based on the most common and standard classifications: the Millennium Ecosystem Assessment classification with its 17 ES divided into four categories, the TEEB with its 22 ES also divided into four categories, and the classification developed by Haines-Young and Potschin [34], the CICES (Common International Classification of Ecosystem Goods and Services), with its nine classes and three themes which excludes the "supporting services categories" and the habitat services/functions. Our selection was also based on our own database. This database was first developed as part of a literature review by Dupras [35] which included studies of ES valuation on forest, wetland and agricultural ecosystems. In the context of this study, the database was supplemented with a literature review of the economic valuation exercises of ES that had been undertaken in Quebec and Ontario [e.g. 9, 18, 36, 37] and with studies on the valuation of aquatic ecosystems. The selected studies from this database are presented in S1 Table. 13 ES were selected for further analysis in the context of the NCC's Green Network. Although the ecosystems under study generate more ES, we limited our analysis to the ES whose value had been estimated previously and to ES for which we could estimate the value using the data available in the allotted time. In order to show the gap between the real contributions of ecosystems to the provision of ES, we use a hollow dot in Table 1 to identify ES provided by ecosystems that were not valued in this study.

## Ecosystem services valuation methods

The analysis of the 13 ES mentioned above was carried out using four methodological approaches. The market pricing method was used to determine the economic value of agricultural products, pollination, and recreational activities and tourism. For all of the other non-market ES, the benefit transfer approach with adjustment, the benefit transfer method with meta-analysis, and the replacement cost method were used. Considering the large number of ecosystems that compose the NCC's Green Network and the quantity of ES that they produced, the use of these methods enabled us to perform the analysis given resource constraints (time, especially), and carry out the main objective of this study. These methods will be discussed in the subsections below.

**Table 1. Selection of ES based on ecosystems and data available.**

| Ecosystem services | Ecosystems / Land uses | | | | | |
|---|---|---|---|---|---|---|
| | Urban forest | Rural forest | Wetlands | Cropland | Prairie and grassland | Freshwater |
| Provisioning services | | | | | | |
| Agricultural services | | | | • | • | |
| Other food | ○ | ○ | ○ | | | ○ |
| Raw material | ○ | ○ | ○ | ○ | ○ | ○ |
| Genetic diversity | ○ | ○ | ○ | ○ | ○ | ○ |
| Regulating services | | | | | | |
| Global climate regulation | • | • | • | ○ | • | ○ |
| Air quality | • | • | ○ | ○ | ○ | ○ |
| Water provisioning | • | • | • | ○ | ○ | ○ |
| Waste treatment | • | • | • | ○ | ○ | • |
| Erosion control | • | • | | • | • | ○ |
| Pollination | • | • | ○ | ○ | • | ○ |
| Habitat for biodiversity | • | • | • | ○ | • | • |
| Disturbance / natural hazards prevention | • | • | • | ○ | ○ | ○ |
| Pest management | • | • | ○ | ○ | ○ | ○ |
| Nutrient cycle | • | • | ○ | • | • | ○ |
| Cultural services | | | | | | |
| Aesthetics (landscape) | ○ | ○ | ○ | • | • | ○ |
| Recreational activities and tourism | • | • | • | • | • | • |
| Scientific and educational | ○ | ○ | ○ | ○ | ○ | ○ |
| Spiritual | ○ | ○ | ○ | ○ | ○ | ○ |
| Heritage and cultural identity | ○ | ○ | ○ | ○ | ○ | ○ |

• ES for which we have values; ○ ES for which we do not have access to values

In the case where we would have had time to calculate a value that is more specific to our study area, we could have used other methods. Indirect valuation methods include stated and revealed preference approaches. Revealed preference approaches are based on observed behavior [38]. They include the travel cost method, where the value of recreation, for example, is estimated as a function of the distance traveled to a location and of the purchases made on the way to and at the location [39]. They also include the hedonic pricing method, where the aesthetics ecosystem service, for example, can be estimated based on the premium associated to the value of a house that is located in a specific landscape [39]. The stated preference approaches are based on stated behavior, generally gathered as part of a survey, where hypothetical scenarios are presented to individuals. They include contingent valuation and choice experiment methods, where respondents are asked, for example, how much they are willing to pay to preserve an aesthetically pleasing landscape (contingent valuation) or are asked to choose their favorite option of a recreational area, where each scenario is composed of a number of given attributes (choice experiment) [40]. For more information on the methods and suggestions about the most appropriate ones to use in a specific context, we suggest consulting the Ecosystem Services Toolkit [41] or the TESSA [42].

**Market pricing method.** The market pricing method estimates the values of tradeable ES by looking at their transaction costs. To perform this analysis for agricultural products, we calculated the economic rents for every crop grown in the NCC's Green Network, noting that most of the agricultural lands were located in the Greenbelt. We first had to determine which crops were grown in the area, then we performed research to determine the value of the

economic rent per crop, which is equal to the revenues generated from the sale of the products minus the total costs of producing them. Economic data for crops came from the Financière agricole du Québec [43–47], the Ontario Ministry of Agriculture, Food and Rural Affairs [48, 49], and the Quebec Reference Centre for Agriculture and Agri-food [50–52].

The economic valuation of the pollination service was based on the method described in Winfree et al. [53], where the value of the pollinating service from wild pollinators, in this case, is based on the value of production. As our focus was on wild pollinators, we used the hypothesis from the study by Rands and Whitney [54] which studies the effect of wild field hedges for nesting pollinators and their travel capacity which varies from 125 m to 2 km. As a conservative distance, we chose to consider the distance of 1 km that pollinators can travel towards agricultural lands from nesting sites. By doing so, we estimated the pollinating potential of wild field hedges (forests and prairies and grassland) on the croplands. This was done by measuring the area in a 1 km buffer around the crops dependent on pollination that contains is made of forests and prairie or grassland (surface$_{FPG}$). The equation we used is the following:

$$V_{pollination} = ((P * Y - C) * D * \rho) * surface_{FPG} \tag{1}$$

Where $V_{pollination}$ is the value of the pollination service, P is the price of the crop, Y is the yield, C are the costs of production, D is the crop dependency on insect pollinators, and $\rho$ is the proportion of insect pollinators [53]. In our case, we used the inverse of the values provided by Morse and Calderone [55]; their value reflected the proportion of pollinators that are honey bees. This value, which represent pollinators that are not honey bees, is 0.1 for berries, 0.5 for soybeans and beans, and 0.4 for forage in rural areas. We estimate that this value ($\rho$) is divided by half in urban areas, considering the quality of the nesting environment [Fetridge et al. 56]. We limited our estimation to crops that have a dependence on pollinators. We used the value of the net benefit (P*Y–C) from beans, berries, soybeans, and forage as calculated for the provisioning service, and took into account their surface area and their rates of dependence on pollinators. We used the degree of dependence as reported by Morse and Calderone [55, 57] for berries/strawberries (0.2), soybeans (0.1), and forage (1). For beans, although the degree of dependence was not provided in Morse and Calderone [55], we assumed a degree of dependence of 0.1, as it is a flowering crop.

The value of recreational activities and tourism is estimated based on the value of user fees charged by the NCC for parking, to access the Gatineau Park and to carry out activities like snowshoeing, skiing, and camping. As the general admission to the Park is free, not all activities carried out in the park are covered by user fees; thus the value is likely underestimated.

**Replacement cost method.** This method is a cost-based approach and estimates the value of an ES, in this case the climate regulation service, based on the cost of the damage that would be induced by its loss. The climate regulation service is evaluated in terms of carbon storage and sequestration, using the value of the social cost of carbon (SCC) as determined by Environment and Climate Change Canada [58]. The SCC is a measure of the expected damage of climate change at a global level resulting from the emission of one additional ton of carbon dioxide ($CO_2$; where 1 ton of C ≈ 3.67 tons of $CO_2$) into the atmosphere over the course of one year. To determine the value of carbon storage and sequestration, a discount rate of 3% was used in the present value calculation, as recommended by Environment and Climate Change Canada [58]. To obtain the value of carbon storage annually, we calculated the present value of carbon stocks over 50 years. To estimate the value of the carbon stored and sequestered by specific features of ecosystems, we relied on estimates from the literature. These specific studies are mentioned in the Results section.

**Benefit transfer with adjustment.** The benefit transfer valuation method uses the results of economic valuations of ES from other sites and transfers the economic values onto the target site. The transfer is performed only when the economic valuation of the transfer site comes from primary data and if the initial site has a similar biophysical environment (e.g. climate) and similar socio-economic characteristics, as represented by the GDP per capita.

To perform this benefit transfer, we created a database that included land classes found in the NCC's Green Network and the ES they produce. We started from a database that was created for the purpose of the Greater Montreal Greenbelt analysis [36], and then included studies gathered from an extensive analysis of the scientific literature using recent and representative studies of the NCC's Green Network. This analysis was undertaken using existing databases such as the Environmental Valuation Reference Inventory (www.evri.ca) and the Ecosystem Services Valuation Database [59], but also by performing research in specialized search engines (e.g. EconLit, Francis, Scopus).

The database that was built contains 149 monetary estimates, all of which came from 78 different studies published between 1990 and 2016 in peer-reviewed journals. The database includes monetary estimates for ES for the five ecosystems under study (forests and woodlands, wetlands, prairies and pastures, croplands, and freshwater). This database is different from a database that can be used for a meta-analysis, because of the number of estimates that is available per ES and per ecosystem, and because of the subsequent use of the monetary estimates. Monetary estimates from this database are adjusted and then directly used to perform benefit transfers. When the values were in a currency other than Canadian dollars, we performed a first adjustment using Purchasing Power Parity (OECD), then we adjusted all values in Canadian dollars using the consumer price index (Bank of Canada).

The level of accuracy of this methodology is based on the level of similarity between the transfer and the target sites. This is why we used two characteristics to limit the number of studies that can be used to undertake the benefit transfer. We first used an ecological criterion, to ensure all transfer sites came from temperate ecosystems and have similarities to the NCC's Green Network. The second criterion was based on socio-economic characteristics and ensured that transfer sites were found in regions and countries that have comparable socio-economic and demographic conditions to the target site. This second criterion is especially important given that the value of ES is related to their contribution to communities' well-being. Based on these criteria, the studies selected as transfer sites mainly came from North America and Western Europe (United States, Canada, Italy, France, Finland, Sweden, Austria, United Kingdom, Ireland).

The last step of this economic valuation method was to perform an adjustment to transform values in Canadian dollars of 2015. The first step was to convert the values in Canadian dollars using purchasing power parity conversion tables (OECD stats), a method more precise than simply using exchange rates, because it takes into account the purchasing power of each currency. Then, the values in Canadian dollars were transferred into 2015 dollars using the proper inflation rates. All of the values used as part of the benefit transfer are described in detail in S1 Table, with the country where the valuation took place, the methodology, the value in $2015 CAD/ha/yr and the ecosystem service(s) studied.

**Benefit transfer with meta-analysis.** The meta-analysis is a statistical method that uses information from a large number of independent studies to infer a value onto a target site, based on characteristics of the system under study. This approach differs from the benefit transfer method with adjustment mainly because, in the latter, the monetary value from one or more primary studies is directly transferred to a target site, with an adjustment for currency and year. In the case of the benefit transfer with meta-analysis, the monetary values from primary studies are not directly transferred onto a target site. Rather, the model transfers the

explanatory factors related to significant socio-economic and environmental characteristics that are associated to the value of an ecosystem. This method reduces transfer errors, thus making it more precise and rigorous than other benefit transfer methods. Despite this increase in precision, a meta-analysis is more complex to undertake and the availability of models is low, because existing models are few and/or privately owned.

The meta-analysis model used as part of this study was developed by He et al. [31] and allowed us to estimate the value of three ES (disturbance prevention, waste treatment and habitat for biodiversity) generated by wetlands. In the initial model, the value of commercial products generated by wetlands is also estimated, but this ES doesn't apply in the context of this study, because there are no commercial activities in wetlands in this area. This specific model cannot be used to estimate the value of ES provided by ecosystems other than wetlands, because the coefficients that represent explanatory variables are defined using a database of studies about wetlands. The explanatory variables and their associated coefficients used as part of this meta-analysis are presented in Table 2. Eq (2) was used to calculate values of the ES. The values of the constant $\hat{a}$ and of the coefficients $\hat{b}var$ had been estimated by the meta-analysis [31].

$$\hat{Y}j = \exp(\hat{a}) * \exp(\hat{b}servXservj) * \exp(\hat{b}wXwj) * \exp(\hat{b}geoXgeoj) * \exp(\hat{b}ecoXecoj)$$
$$* \exp(\hat{b}typeXtypej) \tag{2}$$

In the equation, j refers to each wetland that was assessed. The dependent variable ($\hat{Y}$) is represented by the value of the natural logarithm of the stock of natural capital of 1 ha of wetland.

In order to include site-specific information about the NCC's Green Network in the statistical model, we performed a documentary and a spatial analysis. The documentary analysis

**Table 2. Description of explanatory variables for the meta-analysis (adapted from He et al. [31]).**

| Category | Variable | Coefficient | Summary Description |
|---|---|---|---|
| Wetlands' Ecosystem Services (serv) | Biodiversity Habitat | 1.584 | The wetland holds a particular biodiversity and natural habitat |
| | Waste Treatment | 0.893 | The wetland removes pollutants and filtrates water |
| | Disturbance Prevention | 1.485 | The wetland provides its management role of flood control and retention |
| | Commercial Activities | 1.899 | The wetland allows commercial activities that are either commercial fishing, hunting or ducks breeding |
| Type of Wetland (w) | Manmade | 2.505 | The wetland is not natural (i.e. built by man) |
| | Isolate | -0.856 | The wetland is isolated |
| | Complex | 0.868 | The wetland is complex |
| Geographic Characteristics (geo) | Agriculture | -0.019 | The percentage (%) of the territory in agriculture area within a radius of 10 km around the wetland |
| | Urban | 0.007 | The percentage (%) of the territory in urban area within a radius of 10 km around the wetland |
| | ln wetlands' size | -0.560 | Logarithm of the size of wetlands in hectares |
| Socio-economic Characteristics (eco) | ln GDP per capita | 1.291 | Logarithm of GDP per capita in PPP 2003 USD |
| Type of Value (type) | Marginal | 1.484 | Economic value of wetland was determined for a marginal change. |
| | Median | 3.004 | The economic value of wetland reported in the primary study is a median |
| | Stated preferences | 1.087 | The study is either based on contingent valuation or choice experiment methods |
| Constant (â) | | -3.668 | |

Adapted from Dupras et al. [S1 File]

involved going through land classifications [33] and reports [26, 29] to verify wetland uses and obtain the GDP per capita of people living in the Ottawa-Gatineau region [20–22]. Using this information, we identified values corresponding to categories of ES provided by wetlands and associated them to wetland types and to socio-economic characteristics. The method used to conduct the spatial analysis, an essential step towards identifying values for the categories of geographical characteristics, was based on the methodology developed by He et al. [31]. Using ArcGIS's software, we divided the NCR into sub-regions of 50 km$^2$, and then measured the total size of wetlands and the percentage of agricultural and urban lands surrounding each individual wetland.

## Results

### Spatial analysis

The NCC's Green Network, composed of the Greenbelt, the Gatineau Park and Urban Lands, represent more than 55,000 hectares and about 11% of the total National Capital Region's spatial area. The most important land uses in the NCC's Green Network are forests (72%) and agricultural lands which include pastures/forages and fallow land (10%), followed by urbanized areas (8%) and combined freshwater systems and wetlands (10%). The description of the land cover analysis is presented in Fig 1 and Table 3. When compared to the entire NCR's land cover, forests in the NCC's Green Network are overrepresented (72% vs 49%), and urban and agricultural lands are underrepresented. These differences can be explained by the mandate of the NCC in managing those lands for conservation, recreation and improvement of federal lands.

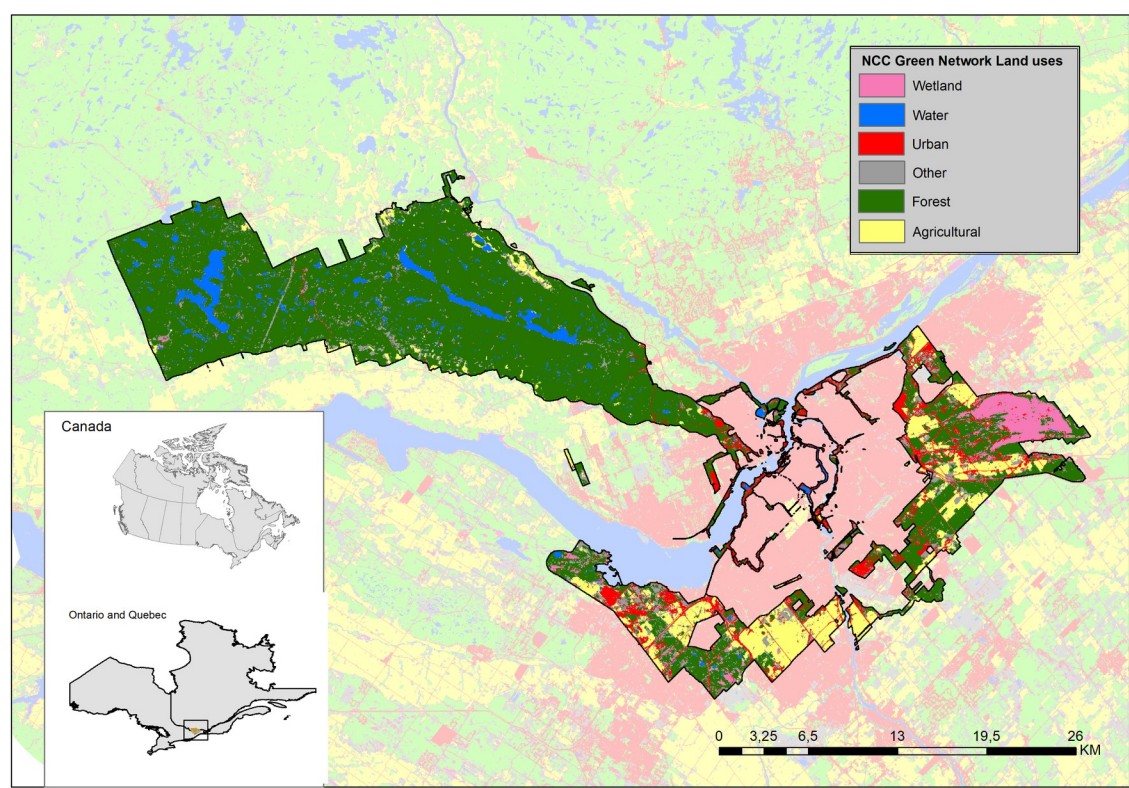

**Fig 1. Land use cover of the Canada's National Capital Green Network.** (adapted from: Dupras et al. [S1 File]).

**Table 3. Land use cover, ecosystem services and valuation of the NCC's Green Network and of the National Capital Region.**

| ES | Area (ha) | Method of valuation | Benefit transfer with Adjustment | | |
|---|---|---|---|---|---|
| | | | Nb of Values | Sources | Methods |
| **Urban Forests** | **1552** | | | | |
| Global climate regulation | | RC | | | |
| Air quality | | BTA | 1 | [60] | i-Tree |
| Water provisioning | | BTA | 3 | [61, 62] | RC, MP |
| Waste treatment | | BTA | 1 | [62] | RC |
| Erosion control | | BTA | 3 | [62, 64] | RC, AC, HP |
| Pollination | | MP | | | |
| Habitat for biodiversity | | BTA | 3 | [65–67] | CV |
| Disturbance prevention | | BTA | 2 | [62] | AC |
| Pest management | | BTA | 1 | [9] | BT |
| Recreation | | MP | | | |
| **Rural forests** | **38128** | | | | |
| Global climate regulation | | RC | | | |
| Air quality | | BTA | 1 | [68] | AC |
| Water provisioning | | BTA | 5 | [61, 62, 69, 71] | RC, MP |
| Waste treatment | | BTA | 4 | [63, 70, 71] | RC, MP |
| Erosion control | | BTA | 6 | [62, 69–71] | RC, OC, HP |
| Pollination | | MP | | | |
| Habitat for biodiversity | | BTA | 14 | [65–67, 72–77] | CE, CV, PF, DC |
| Pest management | | BTA | 2 | [9, 71] | BT, RC |
| Nutrient cycling | | BTA | 3 | [70, 71] | MP, RC |
| Recreation | | MP | | | |
| **Wetlands** | **2453** | | | | |
| Global climate regulation | | RC | | | |
| Water provisioning | | BTA | 2 | [78, 79] | RC |
| Waste treatment | | BTM | | | |
| Habitat for biodiversity | | BTM | | | |
| Disturbance prevention | | BTM | | | |
| Recreation | | MP | | | |
| **Croplands** | **3306** | | | | |
| Food production | | MP | | | |
| Erosion control | | BTA | 3 | [80, 81] | AC, RC |
| Nutrient cycling | | BTA | 1 | [9] | BT |
| Aesthetics | | BTA | 5 | [82–86] | CE, CV, HP |
| Recreation | | BTA | 1 | [9] | MP |
| **Prairie and grasslands** | **2320** | | | | |
| Agricultural products | | MP | | | |
| Global climate regulation | | RC | | | |
| Erosion control | | BTA | 3 | [80, 81] | AC, RC |
| Pollination | | MP | | | |
| Habitat for biodiversity | | BTA | 1 | [9] | BT |
| Pest management | | BTA | 1 | [9] | BT |
| Nutrient cycling | | BTA | 1 | [81] | RC |
| Aesthetics | | BTA | 5 | [82–86] | CE, CV, HP |
| Recreation | | MP | | | |

*(Continued)*

**Table 3.** (Continued)

| ES | Area (ha) | Method of valuation | Benefit transfer with Adjustment | | |
| --- | --- | --- | --- | --- | --- |
| | | | Nb of Values | Sources | Methods |
| **Freshwater** | **1643** | | | | |
| Habitat for biodiversity | | BTA | 1 | [87] | CR |
| Waste treatment | | BTA | 1 | [87] | CR |
| Aesthetics | | BTA | 1 | [87] | CR |
| Recreation | | MP | | | |

AC (Avoided cost), BT (Benefit transfer), CE (Choice experiment), CV (Contingent valuation), CR (Contingent ranking), DC (Damage cost), HP (Hedonic pricing), MP (Market price), OC (Opportunity cost); RC (Replacement cost).

### Ecosystem services valuation

The results of the economic valuation of ES are presented by the type of ecosystem, i.e. forest, wetlands, croplands, grasslands and freshwater systems.

**Forests and woodlands.** ES provided by forests and woodlands vary based on their location, whether they are in an urban, rural or peri-urban setting, and based on their composition, i.e. whether the forest is composed mainly of broadleaves, conifers or is mixed. Based on the spatial analysis, 4% of the forest cover is known to be located in urban area and 96% in rural area. In terms of forest composition, 57% is mixed, 40% is composed mainly of broadleaves, and 3% has a majority of conifers. Forests and woodlands provide 11 ES in the NCC's Green Network.

The ES of global climate regulation was calculated using the replacement cost method and decomposed into carbon sequestration and storage. Carbon sequestration, which represents the annual flow of carbon stored in forests and woodlands, was calculated using the value of the SCC estimated by Environment and Climate Change Canada (ECCC) at $43/tonne of $CO_2$eq [58], and the average value of recorded carbon sequestration rates by ECCC between 1990 and 2013 [82]. This flow corresponds to 1.93 $tCO_2$/ha/year and the economic value of carbon sequestration is valued at $83/ha/year.

Carbon storage, or the stock of carbon stored in forests and woodlands, was evaluated using data from a long-term study undertaken by Kurz and Apps [88]. As part of their analysis, they evaluated carbon fluxes in the Canadian forest sector over a 70-year period and estimated that the stock of carbon stored in cool temperate forests was equal to 220 tonnes of carbon per hectare or 807 tonnes of $CO_2$eq per hectare. Over the forested surface, it means that nearly 9 million tonnes of carbon are stored in the NCC's Green Network.

To obtain this value annually, we calculated the present value of carbon stocks over 50 years, at a 3% discount rate. Using these parameters, the annual value for carbon storage was evaluated at $158/ha/year. Overall, the economic value for climate regulation is $241/ha/year.

Air quality was calculated using the benefit transfer method, more specifically using Hirabayashi's [60] I-tree software, which has the ability to differentiate between urban and rural effects of trees on air quality. The multipliers used in the analysis are provided in Table 4. The resulting value for the ES of air quality is $10/ha/year for rural forests and $554/ha/year for urban forests.

Pollination was calculated using the market price method as described in the methodology. The value of this service is estimated at $10/ha/year for rural forests and at $14/ha/year for urban forests in the NCC's Green Network. While calculating the value of the pollinating service, we took into account the quality of the nesting environment for wild pollinators, as it is

**Table 4. Multipliers derived from the United States' total values.**

| Pollutant | Removal Multiplier (kg/ha/yr) | | Value Multiplier ($/ha/yr) | |
|---|---|---|---|---|
| | Urban | Rural | Urban | Rural |
| CO | 1.27 | 1.00 | 1.7 | 0.02 |
| $NO_2$ | 7.00 | 5.45 | 3.1 | 0.04 |
| $O_3$ | 54.04 | 54.93 | 154.1 | 2.6 |
| $PM_{10}$ | 15.34 | 18.51 | 97.3 | 2.1 |
| $PM_{2.5}$ | 2.76 | 2.66 | 297.4 | 4.9 |
| $SO_2$ | 3.44 | 3.47 | 0.5 | 0.01 |
| **Total** | | | **554.1** | **9.7** |

Source: Adapted from Nowak et al. [89] and Hirabayashi [60]; *Adapted from Dupras et al. [S1 File]*

preferable to take into account the quality of the nesting environment, as well as the quality of the agricultural environment [90]. This meant dividing the value of ρ by half for urban crops, according to insights about the quality of urban environments for wild pollinators [56]. The values were obtained by using the sum of the value of pollination for crops grown in rural areas (from Eq 1), divided by the total area in the 1 km buffer around crops that can be considered as nesting habitats for wild pollinators and then multiplied by the share of rural forests in the 1 km buffer (Eq 3). We then did the same for urban forests, rural and urban prairies and grasslands.

$$\text{Pollination ES} = (V_{\text{pollination}}\text{Rural Total}/\text{Area}_{\text{Rural buffer}}) * (\text{Area}_{\text{rural forests buffer}}/\text{Area}_{\text{forest buffer}}) \; (3)$$

The economic value associated to recreational activities and tourism was estimated based on results from the NCC's 2014–2015 Annual Report [91] which showed that the NCC collected $2.7 million in user fees, which is equivalent to $75/ha/year when the $2.7 million value is divided over the Gatineau Park's entire spatial area (36,161 ha). These user fees correspond to annual passes used to access the park during the winter to perform cross-country skiing and snowshoeing, to camping fees and to parking fees within specific areas of the park. When we compare this value to a report by Environics [27], which showed that Gatineau park visitors (77% of residents and 23% of non-residents) spent $184 million in the region as part of their travel to the Gatineau Park in food, sport supply and other purchases such as gas, the $2.7 million in user fees seems quite low. As a result, the value of $75/ha/year is most likely insufficient to represent the real value of the recreational potential of the Gatineau Park. It is nevertheless the only value that we can directly tie to activities performed in the Park.

The remaining seven ES for forests and woodlands are evaluated based on the benefit transfer method and are described below. The primary studies which were used to perform the benefit transfer with adjustment are provided in S1 Table.

Habitat for biodiversity is part of the ES with the highest value per hectare for forests and woodlands. The value of this ES has been estimated using three monetary estimates for urban forests [65–67], resulting in a mean value of $2,684/ha/year, and using fourteen estimates for rural forests [65–67, 72–77], for a mean value of $2,185/ha/year.

The service of disturbance prevention, or flood control, is estimated only for urban forests, because of its impact on the stormwater control and retention. In Morri et al. [62], this service is calculated based on the cost of a water reservoir. Using two monetary estimates [62], this service has a mean value of $5,030/ha/year, making it the ES with the highest value per hectare for forests and woodlands.

Water provisioning was estimated using three values for urban forests [61, 62] and four values for rural forests [61, 62, 69, 71], which rendered a mean value per hectare of, respectively, $339/ha/year and $839/ha/year.

The ES of nutrient cycling has been calculated only for rural forests, using three monetary estimates [70, 71], and has an estimated value of $319/ha/year. The values from the benefit transfer come from studies that have used the market pricing method [70] and the replacement cost method [71].

Waste treatment refers to the capacity of forests and woodlands to prevent and reduce the proportion of nutrients and pollutants from going directly into rivers and to their ability to filter, store and transform pollutants. This ES is estimated at $140/ha/year for urban forests [62], and at $318/ha/year for rural forests [63, 70, 71].

Erosion control has an estimated value of $211/ha/year for urban forests [62–64] and of $137/ha/year for rural forests [62, 69–71], and pest management was estimated at $45/ha/year for urban forests [9] and at $30/ha/year for rural forests [9, 71].

Table 5 presents each monetary estimate per ES for urban and rural forests. This table also includes minimum, maximum and mean values, as well as the standard deviation for ES that

**Table 5. Value of ES provided by forests and woodlands of the NCC's Green Network ($CAD 2015).**

| Ecosystem Services | Nb. of $ estimates | Total area (ha) | Min. value ($/ha/y) | Max. value ($/ha/y) | Mean ($/ha/y) | Std. dev. ($/ha/y) | Method | Total value ($'000/y) |
|---|---|---|---|---|---|---|---|---|
| **Urban Forests** | **15** | **1552** | **6797** | **14,312** | **9328** | | | **14,484.8** |
| Global Climate Regulation | 1 | | - | - | 241 | nd | RC | 374.0 |
| Air Quality | 1 | | - | - | 554 | nd | BT | 859.8 |
| Water Provisioning | 3 | | 203 | 609 | 339 | 234 | BT | 526.1 |
| Waste Treatment | 1 | | - | - | 140 | nd | BT | 217.3 |
| Erosion Control | 3 | | 111 | 396 | 211 | 161 | BT | 327.5 |
| Pollination | 1 | 1582[a] | - | - | 14 | nd | MP | 22.1 |
| Biodiversity Habitat | 3 | | 444 | 7160 | 2684 | 3876 | BT | 4,165.6 |
| Disturbance Prevention | 2 | | 4975 | 5085 | 5030 | 78 | BT | 7,806.6 |
| Pest Management | 1 | | - | - | 45 | nd | BT | 69.8 |
| Nutrient Cycling | - | | - | - | - | - | - | - |
| Recreation | 1 | | - | - | 75 | nd | MP | 116.4 |
| **Rural Forests** | **36** | **38,128** | **498** | **16,971** | **4161** | | | **158,529.1** |
| Global Climate Regulation | 1 | | - | - | 241 | nd | RC | 9188.9 |
| Air Quality | 1 | | - | - | 10 | nd | BT | 381.3 |
| Water Provisioning | 5 | | 123 | 3053 | 839 | 1252 | BT | 31,989.4 |
| Waste Treatment | 4 | | 26 | 806 | 318 | 344 | BT | 12,124.7 |
| Erosion Control | 6 | | 1 | 536 | 137 | 202 | BT | 5,223.5 |
| Pollination | 1 | 16,501 | - | - | 10 | nd | MP | 165.0 |
| Biodiversity Habitat | 14 | | 0.1 | 11,349 | 2185 | 3673 | BT | 83,309.7 |
| Disturbance Prevention | - | | - | - | - | - | - | - |
| Pest Management | 2 | | 14 | 45 | 29 | 22 | BT | 1124.8 |
| Nutrient Cycling | 3 | | 0.1 | 848 | 319 | 462 | BT | 12,162.8 |
| Recreation | 1 | | - | - | 75 | nd | MP | 2859.6 |

[a]: The area of urban forests is larger than the total as it includes some forests that are classified as "Rural" in our database. Std. dev.: Standard deviation; BT: Benefit Transfer; MP: Market Price; RC: Replacement Cost; *Adapted from Dupras et al.* [S1 File]

have been estimated using the benefit transfer method and at least two monetary estimates. As indicated in the table, when using mean values for urban and rural forests, the total value for forest ecosystems is equal to $173.0 million per year. When taking into account the forest cover, the most important ES in terms of economic value are habitat for biodiversity, water provisioning, and nutrient cycling.

**Wetlands.**   Wetlands represent 4.5% of the NCC's Green Network landscape. Most wetlands are relatively small, except the Mer Bleue bog, which is a 3,500-hectare conservation area.

The meta-analysis approach was used to determine the value of three ES: habitat for biodiversity, waste treatment and disturbance prevention. Different criteria were included as part of this statistical method, namely wetland size and type (man-made, isolate, complex), GDP per capita, and land cover composition around the wetlands (e.g. % agriculture, % urbanized areas). After performing the analysis for each wetland in the study area, we found a value of $21,461/ha/year for habitat for biodiversity, $15,893/ha/year for waste treatment, and $20,766/ha/year for disturbance prevention. As mentioned, no value was found for market services, because there are no commercial activities carried out in wetlands in the study area.

As a way of comparison, we found a number of studies that have used other methods to estimate the value of wetlands. For urban wetlands, the service of habitat was estimated by Kosz [92] in Austria at a value of $149,161/ha/year to $338,960$/ha/year using contingent valuation. For rural wetlands, the value ranges from $22/ha/year [93] to $4,251/ha/year [94] when using contingent valuation methods. For the service of waste treatment, urban wetlands have an estimated value of $0.3/ha/year [79] to $19,713/ha/year [95], using the avoided costs method. Rural wetlands have an estimated value that ranges from $1,697/ha/year [79] to $6,282/ha/year [96] using the replacement cost method. Finally, the disturbance prevention service has been estimated, for both rural and urban wetlands, at a value ranging from $77/ha/year [97] using the market price method to $5,967/ha/year [98] using the avoided costs method. The high values obtained with the meta-analysis, when compared to the results from the studies mentioned above, are largely explained by two environmental factors: the scarcity of wetlands, as it increases the relative importance of the ES provided, and their geographic location, especially with regards to the proximity to agricultural and urban lands which affects the importance of the ES provided. These values are also explained by socio-economic factors, namely demographics and the relative wealth of the population, as these factors influence the value of the ES.

As carbon sinks, wetlands provide global climate regulation services. Using the values of carbon stocks found in peat bogs across representative regions in the province of Quebec [99], an average stock of 1,468 tonnes of carbon per hectare was found for the NCC's Green Network wetlands. An economic value for carbon storage of $1,057/ha/year was found using the value of the SCC ($43/tonne of $CO_2$), a 50-year annualization period, and a 3% discount rate.

The carbon sequestration value was estimated based on the Mer Bleue wetland sequestration rate of 0.7 tC/ha/year [100], and rendered an economic value of $111/ha/year. By combining these two values, the global climate regulation service is estimated at $1,168/ha/year.

The value of recreational services of $75/ha/year was estimated based on the Gatineau Park analysis. The use of this value is based on the fact that accessible wetlands and their surrounding areas in the NCC Green Network are mainly used for birdwatching, biking, hiking, snowshoeing and cross-country skiing. Shirley's Bay, in the Ottawa Greenbelt, is the only place where ice fishing is allowed. These are activities that some people pay to access in the Gatineau Park. It is also difficult to estimate the value of the recreational service in wetland areas specifically, due to the limited number of studies and the fact that all of the activities in the Greenbelt, where the largest wetlands are located, are free. In addition, most studies that evaluate the

**Table 6. Value of ES provided by wetlands of the NCC's Green Network ($CAD 2015).**

| Ecosystem Services | Nb. of $ estimates | Total area (ha) | Min. value ($/ha/y) | Max. value ($/ha/y) | Mean ($/ha/y) | Std. dev. ($/ha/y) | Method | Total value ($'000/y) |
|---|---|---|---|---|---|---|---|---|
| **Urban and Rural Wetlands** | **7** | **2453** | **59,371** | **59,417** | **59,394** | | | **145,693.5** |
| Global Climate Regulation | 1 | | - | - | 1168 | nd | RC | 2865.1 |
| Water Provisioning | 2 | | 8 | 54 | 31 | 32 | BT | 76.0 |
| Waste Treatment | 1 | | - | - | 15,893 | nd | BT | 38,985.5 |
| Biodiversity Habitat | 1 | | - | - | 21,461 | nd | BT | 52,643.8 |
| Disturbance Prevention | 1 | | - | - | 20,766 | nd | BT | 50,939.0 |
| Recreation | 1 | | - | - | 75 | nd | MP | 184.0 |

Std. dev.: Standard deviation; BT: Benefit Transfer; MP: Market Price; RC: Replacement Cost; *Adapted from Dupras et al. [S1 File]*

recreational service of wetland include fishing and hunting as part of their activities. In this case, it would be inappropriate to compare our value to these, as hunting and fishing are generally not allowed in the study area.

The last service to be estimated is water provisioning. Its value of $31/ha/year was determined based on two monetary estimates from the benefit transfer database [78, 79].

The combined value of all six services provided by wetlands is equal to $59,394/ha/year and a total value of $145.7 million/year (Table 6).

**Croplands.** Croplands represent 10% of the NCC's Green Network land cover, and the main agricultural crops grown in the Greenbelt are soy, corn and barley.

The value of the agricultural production service was evaluated using the market pricing method. The net benefit was calculated for barley, oat, wheat, corn, soy, dry beans, strawberries, and other cereals, and rendered a mean value of $919/ha/year or $3 million per year.

The value of recreational activities was estimated based on a study by Dupras and Alam [9], where the income from agro-tourism of 66 agro-businesses in the Greater Montreal area was estimated at $94/ha/year.

The other non-market ES were estimated using the benefit transfer with adjustment method. Using the benefit transfer database, a value of $112/ha/year was identified for erosion control [80, 81], of $184/ha/year for nutrient cycling [9], and of $76/ha/year for landscape aesthetics [82–86].

The total mean value of all these ES is equals $4.59 million per year, with an average of $1,389/ha/year (Table 7).

**Table 7. Value of ES provided by croplands of the NCC's Green Network ($CAD 2015).**

| Ecosystem Services | Nb. of $ estimates | Total area (ha) | Min. value ($/ha/y) | Max. value ($/ha/y) | Mean ($/ha/y) | Std. dev. ($/ha/y) | Method | Total value ($'000/y) |
|---|---|---|---|---|---|---|---|---|
| **Croplands (Annual Crops)** | **11** | **3306** | **1279** | **1581** | **1389** | | | **4592.0** |
| Food Production | 1 | | - | - | 919 | nd | MP | 3038.2 |
| Erosion Control | 3 | | 61 | 193 | 109 | 73 | BT | 360.3 |
| Nutrient Cycling | 1 | | - | | 184 | nd | BT | 608.3 |
| Aesthetics | 5 | | 21 | 191 | 83 | 76 | BT | 274.4 |
| Recreation | 1 | | - | - | 94 | nd | BT | 310.8 |

Std. dev.: Standard deviation; BT: Benefit Transfer; MP: Market Price; *Adapted from Dupras et al. [S1 File]*

**Prairies, pastures and grasslands.** Prairies, pastures and grasslands are represented by pasture and forage lands, and by fallow lands.

In addition to the value of agricultural products in the form of forages, which have a value of $116/ha/year [52], these ecosystems provide many ES that are incremental to those provided by croplands, because of the positive effects of non-intensive land use.

As opposed to croplands, a value was calculated for the climate regulation services provided by grasslands, pasture and forage crops. Smith et al. [101] estimated that these ecosystems contain on average 105 tonnes of carbon per hectare. The value of carbon storage, annualized over 50 years, with a discount rate of 3% and the value of the SCC ($43/tonne of $CO_2$) is equal to $76/ha/year. The value of carbon sequestration, using a sequestration rate of 2.17 tonnes of C/ha, is equal to $342/ha/year. The total value for climate regulation is then of $418/ha/year.

Recreational services have been estimated at $75/ha/year, based on the results obtained for the Gatineau Park. The pollination service value was calculated for urban and rural prairies and grasslands, using the same methodology as the one used for forests and woodlands. However, due to the very small land area from this land use present in buffers around agricultural areas, the value we obtained was too small to report. Few studies estimate the value of the recreational service for prairies and grasslands, as a result we will use the reported value from Costanza et al.'s [102], based on the study by Boxall [103], as a comparison for recreation ($4/ha/year).

The remaining five ES estimated for prairies, pastures and grasslands were evaluated using the benefit transfer database. The calculated mean value are $109/ha/year for erosion control [80, 81], $2,467/ha/year for biodiversity habitat [9], $45/ha/year for pest management [9], $147/ha/year for nutrient cycling [81], and $76/ha/year for landscape aesthetics [82–86].

The total mean value for all nine estimated service is $8.0 million per year, or $3,460/ha/year (Table 8).

**Freshwater.** Freshwater systems represent 5% of the NCC's Green Network in the form of lakes, streams and of two rivers. Despite the importance of freshwater systems in the world, few studies have been produced on ES, aside for cultural services. This limited the analysis of freshwater ES to four ES. The service of recreational activities and tourism was evaluated based on the analysis conducted for the Gatineau Park, for a value of $75/ha/year. This value only takes into account the recreational visits to freshwater ecosystems in areas actively managed by the NCC. It does not take into account the fish harvested as part of fishing activities.

**Table 8. Value of ES provided by prairies, pastures and grasslands of the NCC's Green Network ($CAD 2015).**

| Ecosystem Services | Nb. of $ estimates | Total area (ha) | Min. value ($/ha/y) | Max. value ($/ha/y) | Mean ($/ha/y) | Std. dev. ($/ha/y) | Method | Total value ($'000/y) |
|---|---|---|---|---|---|---|---|---|
| **Pastures and Grasslands** | **13** | **2320** | **3350** | **3652** | **3460** | | | **8027.2** |
| Agricultural Products | 1 | | - | - | 116 | nd | MP | 269.1 |
| Global Climate Regulation | - | | - | - | 418 | - | RC | 969.8 |
| Erosion Control | 3 | | 61 | 193 | 109 | 73 | BT | 252.9 |
| Biodiversity Habitat | 1 | | - | - | 2467 | nd | BT | 5723.4 |
| Pest Management | 1 | | - | - | 45 | nd | BT | 104.4 |
| Nutrient Cycling | 1 | | - | - | 147 | nd | BT | 341.0 |
| Aesthetics | 5 | | 21 | 191 | 83 | 76 | BT | 176.3 |
| Recreation | 1 | | - | - | 75 | nd | MP | 174.0 |

Std. dev.: Standard deviation; BT: Benefit Transfer; MP: Market Price; RC: Replacement Cost; *Adapted from Dupras et al.* [*S1 File*]

**Table 9. Value of ES provided by freshwater systems of the NCC's Green Network ($CAD 2015).**

| Ecosystem Services | Nb. of $ estimates | Total area (ha) | Min. value ($/ha/y) | Max. value ($/ha/y) | Mean ($/ha/y) | Std. dev. ($/ha/y) | Method | Total value ($'000/y) |
|---|---|---|---|---|---|---|---|---|
| **Aquatic Systems** | **4** | **1643** | **137** | **137** | **137** | | | **225.1** |
| Biodiversity Habitat | 1 | | - | - | 10 | nd | BT | 16.4 |
| Waste Treatment | 1 | | - | - | 48 | nd | BT | 78.9 |
| Aesthetics | 1 | | - | - | 4 | nd | BT | 6.6 |
| Recreation | 1 | | - | - | 75 | nd | MP | 123.2 |

Std. dev.: Standard deviation; BT: Benefit Transfer; MP: Market Price; *Adapted from Dupras et al. [S1 File]*

The estimated value for biodiversity habitat, waste treatment and aesthetics was adapted from a study by Poder et al. [87] on the willingness to pay for the Blue Network of the Greater Montreal area, where a stated preference method was used. The value of the WTP from the initial study, expressed in dollars per household, was multiplied by the number of households in the Ottawa-Gatineau Metropolitan region, and then this value was divided by the number of hectares of freshwater systems in the region. The details of this adaptation is provided in S2 Table. The annual present value was then calculated using a 20-year actualization period and a 3% discount rate. The values obtained were $10/ha/year for biodiversity habitat, $48/ha/year for waste treatment, and $4/ha/year for aesthetics. These values were chosen because of the proximity of the study area to the Green Network. A number of other studies performed on freshwater ES focus on the issue of eutrophication, the costs of improving water quality [104–108] and the impact of invasive species on property prices [109–111]. Although these issues are present in the Green Network, the aim of our exercise was not to investigate the resolution of a specific environmental issue.

The total value for freshwater systems is $225,091 per year or $137/ha/year (Table 9).

**Total economic value.** The total mean economic values per ES and per ecosystem are presented in Table 10. In dollars per hectare, wetlands are the ecosystems with the highest

**Table 10. Total annual values per hectare for ES of the NCC's Green Network ($CAD 2015/ha/year).**

| Ecosystem Services | Urban Forests | Rural Forests | Wetlands | Croplands | Prairies, grasslands | Freshwater Systems |
|---|---|---|---|---|---|---|
| Agricultural Products | - | | - | 919 | 116 | - |
| Global Climate Regulation | 241 | 241 | 1,168 | - | 418 | - |
| Air Quality | 554 | 10 | - | - | - | - |
| Water Provisioning | 339 | 839 | 31 | - | - | - |
| Waste Treatment | 140 | 318 | 15,893 | - | - | 48 |
| Erosion Control | 211 | 137 | - | 109 | 109 | - |
| Pollination | 14 | 10 | | | - | |
| Biodiversity Habitat | 2,684 | 2,185 | 21,461 | - | 2,467 | 10 |
| Disturbance Prevention | 5,030 | - | 20,766 | - | - | - |
| Pest Management | 45 | 29 | - | - | 45 | - |
| Nutrient Cycling | - | 319 | - | 184 | 147 | - |
| Aesthetics | - | - | - | 83 | 83 | 4 |
| Recreation | 75 | 75 | 75 | 94 | 75 | 75 |
| TOTAL | **9,333** | **4,163** | **59,394** | **1,389** | **3,460** | **137** |
| Number of ha | **1,552** | **38,128** | **2,453** | **3,306** | **2,320** | **1,643** |

Adapted from Dupras et al. [S1 File]

**Table 11. Total annual values ($ '000 CAD 2015) for ES of the NCC's Green Network.**

| Ecosystem Services | Urban Forests | Rural Forests | Wetlands | Croplands | Prairies, grasslands | Freshwater Systems | Total |
|---|---|---|---|---|---|---|---|
| Agricultural Products | - | | - | 3038.2 | 269.1 | - | 3,307.3 |
| Global Climate Regulation | 374.0 | 9,188.9 | 2,865.1 | - | 969.8 | - | 13,397.7 |
| Air Quality | 859.8 | 381.3 | - | - | - | - | 1,241.1 |
| Water Provisioning | 526.1 | 31,989.4 | 76.0 | - | - | - | 32,591.5 |
| Waste Treatment | 217.3 | 12,124.7 | 38,985.5 | - | - | 78.9 | 51,406.4 |
| Erosion Control | 327.5 | 5,223.5 | - | 360.4 | 252.9 | - | 6,164.3 |
| Pollination | 22.1 | 165.0 | - | - | - | | 187.1 |
| Biodiversity Habitat | 4,165.6 | 83,309.7 | 52,643.8 | - | 5,723.4 | 16.4 | 145,859.0 |
| Disturbance Prevention | 7,806.6 | - | 50,939.0 | - | - | - | 58,745.6 |
| Pest Management | 69.8 | 1,124.8 | - | - | 104.4 | - | 1,299.0 |
| Nutrient Cycling | - | 12,162.8 | - | 608.3 | 341.0 | - | 13,112.1 |
| Aesthetics | - | - | - | 274.4 | 192.6 | 6.6 | 473.6 |
| Recreation | 116.4 | 2,859.6 | 184.0 | 310.8 | 174 | 123.2 | 3,768.0 |
| TOTAL | **14,485.2** | **158,529.1** | **145,693.5** | **4592.0** | **8027.2** | **225.1** | **331,552.7** |

Adapted from Dupras et al. [S1 File]

economic value, followed by urban and rural forests. When taking into account land cover (number of ha) associated with each ecosystem as shown in Table 11, rural forests are the ecosystem with the highest economic value. When looking at individual ES, recreation has the highest value ($154 million) followed by biodiversity habitat ($146 million), which is provided by five out of six ecosystems studied. Fig 2 provides a spatial representation of the natural capital value in $/ha/year.

The total mean economic value for the entire NCC's Green Network is $332 million per year, with a minimum of $187 million per year and a maximum of $829 million per year (Table 12). The minimum and maximum values come from benefit transfer estimates, where at least two monetary values were used. The largest variation between minimum and maximum values comes from the rural forest ecosystem, especially from the biodiversity habitat ES, where the standard deviation has a value of $3,673/ha/year and values ranging from $0.1/ha/year to $11,349/ha/year.

Aside from ES values calculated for forests and woodlands ecosystems and the pollination service for prairies and grasslands, values were not differentiated between urban and rural environments, mainly because of data availability. In the case of wetlands, distance to urban areas was taken into consideration in the meta-analysis to define the value, but wetlands were not explicitly labelled as urban or rural in the tables. This is because of the nature of the benefit transfer with meta-analysis which takes into consideration the characteristics of ecosystems in the value estimation. Another reason for not explicitly labelling ecosystems other than forests as urban or rural is the fact that most areas were considered peri-urban. It is especially true in the case of the Greenbelt, because it is enclosed between urban environments. This is also the case for the southern portion of the Gatineau Park.

## Discussion

Currently, the NCC's Green Network has an estimated value of 332 million dollars per year, which represents a mean value of $6,014/ha/year, for a total economic value of more than $5 billion annualized over 20 years. This value represents the contribution of ES provided by forests and woodlands, wetlands, agricultural areas, prairies, grasslands, and pastures, and by

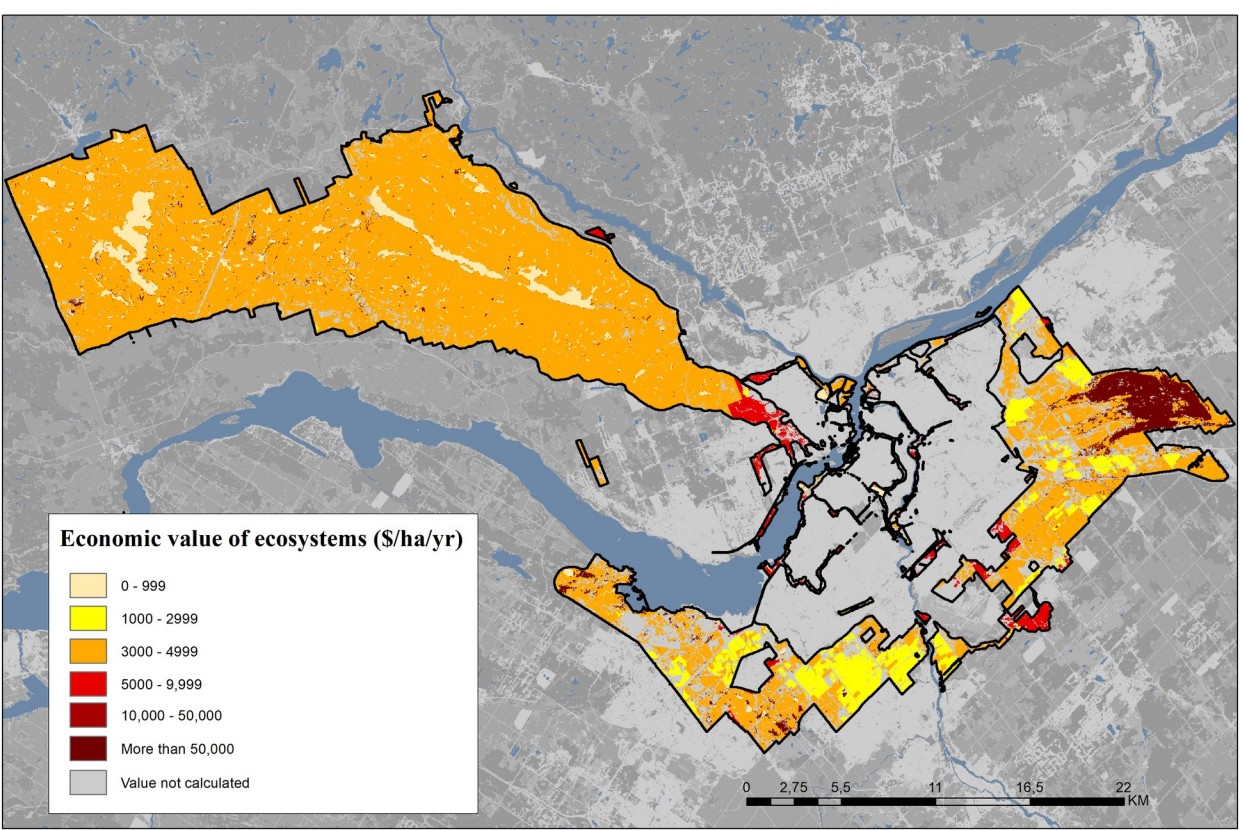

**Fig 2. ES value of Canada's National Capital Green Network (values expressed in $CAD 2015 /ha/year).** (adapted from: Dupras et al. [S1 File]).

freshwater systems. The most important part of this value is represented by nonmarket ES, mainly regulating services, such as biodiversity habitat, waste treatment, disturbance prevention, and global climate regulation.

The aim of this study was in part to explore the value of a suite of regulating services, since in many studies, these services are often left aside because of their difficult association with economic indicators. The interest of exploring the regulating services is all the more relevant as they largely explain the variations in the total economic value for the different ecosystems (Table 10). 76% of the total economic value per ha in the NCC's Green Network is captured by the wetlands, because of their particularly important contribution to habitat biodiversity,

**Table 12. Total minimum, mean, and maximum annual values ($M CAD 2015) for the studied ecosystems.**

| Ecosystems | Minimum value | Mean value | Maximum value |
|---|---|---|---|
| Urban Forests | 10.6 | 14.5 | 22.2 |
| Rural Forests | 19.1 | 158.5 | 647.1 |
| Wetlands | 145.6 | 145.7 | 145.7 |
| Croplands | 4.2 | 4.6 | 5.2 |
| Prairies, pastures, grasslands | 7.8 | 8.0 | 8.5 |
| Freshwater Systems | 0.2 | 0.2 | 0.2 |
| TOTAL | **187.5** | **331.6** | **829.0** |

Adapted from Dupras et al. [S1 File]; $M: '000 000

disturbance prevention, waste treatment and global climate regulation. This result is not surprising since the great importance of the wetland's ecological functions is already demonstrated in the scientific literature [102, 112, 113]. The variation of the total economic value between ecosystems can be explained along a rural-urban gradient. Indeed, at the NCC's Green Network scale, rural forests have a total annual value more than 10 times greater than that of urban forests (Table 11). However, as the area of urban forests is relatively small compared to the vast area of rural forests, the total annual value per hectare of urban forests is more than twice that of rural forests (Table 10). This disparity was also highlighted in the study of Nikodinoska et al. [114] in which only 1% of the economic value was generated by green urban areas of Uppsala (Sweden); when considering the size of different land uses, the average economic value of green urban areas was the highest (53%). These results support the importance of urban ecosystems for the population as they provide an important contribution in terms of regulating and cultural ES. Thus, ES hotspots tended to be located in urban settings where a higher population density led to a stronger demand for ES (see also [115]). Urban-related pressures like air pollution also increase the need for ES, although this can lead to local mismatches between ES supply and demand [116, 117]. Indeed, these results should not obscure the fact that most of the total economic value is provided by the rural ecosystems (Table 11) and that a conservation effort must be made on these ecosystems to promote sustainability. A better understanding of the processes that make up nonmarket and rural ecosystems is important to make more informed decisions on their management and preservation.

The integration of knowledge about the value of ES in decision-making for planning is a relevant and evolving area of study [118–122]. Although there are still few real-world examples of the use of ES knowledge in public planning and decisions [123–125], a few trends seem to emerge relative to their use based on current examples from the literature. The first approach is based on ecosystem-wide planning used to resolve specific issues. Examples of this planning approach is the one adopted by the city of New York to protect its water supply [17, 126], and the one adopted for the management of nitrogen in the Chesapeake Bay in the United States [118]. Another approach is through regional and strategic planning (e.g. [115]), where ecosystems are managed regardless of jurisdictional limits. This includes the planning approach adopted by the United States Forest Service as part of its 2012 Planning Rule, where it is required to plan forest management activities at the ecosystem level. The region of Tampere, in Finland, has also included this approach in its strategic plan towards 2040. The spatial valuation of ES by Tammi et al. [115] provided input into the regional planning strategy by performing an analysis of ecosystems along a urban-rural gradient, but without consideration for inner jurisdictional boundaries. Their results were used by decision makers to plan for ES hotspots. This information also interested local groups within and outside of the Tampere region.

Tools that are not solely based on land use planning also exist to enable the maximization and protection of ES supply. Payments for Ecosystem Services (PES) are such an example. The implementation of PES vary across the world, but they are programs where landowners are compensated (in money or in-kind) for the provision of ES when they carry out agreed upon land use management practices. These programs exist in developing and industrialized countries alike, although PES programs implemented in latter countries tend to focus on agricultural lands [127, 128].

Despite these examples of the use of ES knowledge into policies, plans or tools, the use of ES in decision-making is currently limited by the amount of knowledge available. For example, there are few sets of longitudinal data on ES supply, which limits the tracking of services over a longer period [118, 119]. Other limits include the type of policies and the legislative framework in place in a region, the interest of planners and decision makers, as well as social and political will [115, 119, 122].

In addition, to ensure the provision of ES, planners and decision makers need to understand that the quality and stability of ES are related to land uses, to the capacity of species to interact with one another, and to move across the landscape [129, 130]. As a result, the NCC should facilitate the creation of corridors to increase connectivity within the Green Network and with surrounding natural environments, through land acquisition or partnerships with public and/or private owners [131]. The results of this study can be used to raise awareness on the value of ES, but this new knowledge that is available to decision makers of the region can be taken further. These results can be used as a stepping stone to improve ecosystem-wide planning in the Ottawa-Gatineau region, especially since the report was received with great interest by the NCC, media [132–134], and the community within the immediate Ottawa-Gatineau region, but also within the larger national capital region of Canada.

These results should nevertheless be considered in light of methodological limitations inherent to the approach chosen in this study, especially with regard to the spatial analysis and the economic valuation. Data availability was a limiting factor in both the spatial analysis and the economic valuation. Although tests were made to detect errors in the GIS layer, the chosen source only had an 85% precision. Data availability also impacted the economic valuation, especially when using the benefit transfer approach. Even though care was taken to avoid transfer bias, the results are not as robust as a study based exclusively on primary data. Additionally, the benefit transfer analysis was dependent on published literature, for which the coverage of ecosystems and ES is unequal. This unequal attention affects the accuracy of some monetary estimates. For example, there were more studies on forests and wetlands than on freshwater and agricultural lands. Finally, our analysis was not exhaustive with regard to the valuation of the stock of natural capital, where only the values for Wetlands ES included the stock of natural capital and the values for the service of Global climate regulation included the value of the stock of carbon. This suggests that our estimate of the value of the NCC's Green Network is an underestimation.

Nonetheless, since the objective of the study was not to establish a precise value that would be used to put in place a program or market, but rather to increase the awareness of a number of stakeholders, citizens and decision makers about the real contribution of ecosystems to their well-being, the choice of data and methodology is valid.

## Conclusion

Since the implementation of the first Canadian Greenbelt in Ottawa in 1950, there has been an evolution in the design of greenbelts globally. Instead of surrounding a city to prevent growth, the focus is on building corridors for biodiversity, so that species can move within the landscape and people can enjoy cultural services within cities. The final design of the Golden Horseshoe in Toronto and of the proposed Montreal greenbelt and its associated natural infrastructure network are based on this approach.

Initially, the Plan Gréber of 1950 included provisions to connect the Greenbelt and the Gatineau Park through urban lands. The appeal of connected structures is the ability to improve the production of ES. In the case of the NCC's Green Network, there is an opportunity to connect these landscapes, but it will require discussions between the NCC and municipalities for the purchase and protection of lands. As mentioned in the introduction, an important challenge to the integrity of the green network is encroachment, especially since the green network does not have a formally protected status.

The lack of information on the value of natural habitats to human well-being in urban and peri-urban setting does lead to the degradation of natural capital. With the monetary values elicited as part of this study, managers at the NCC will be able to showcase the contribution of

natural landscapes to reduce the cost of grey infrastructure and to improve the quality of life of citizens.

## Supporting information

**S1 Table. List of studies used as part of the ES valuation with benefit transfer (Values in $2015 CAD).**
(DOCX)

**S2 Table. Conversion of the WTP values from Poder et al. [87] to values for the NCC Green Network.**
(DOCX)

**S1 File. Dupras et al. 2016. natural capital.** The Economic Value of the National Capital Commission's Green Network. Dupras J, L'Ecuyer-Sauvageau C, Auclair J, He J, Poder T. Natural Capital. The Economic Value of the National Capital Commission's Green Network. David Suzuki Foundation & National Capital Commission. 2016. 51 pages.
(PDF)

## Acknowledgments

The authors would like to thank Catherine Verreault, Christie Spence, Matthew Tomlinson, and Franck Fetue Ndefo for their fruitful comments and collaboration in the development of the study.

## Author Contributions

**Conceptualization:** Jérôme Dupras.

**Formal analysis:** Chloé L'Ecuyer-Sauvageau, Jérôme Dupras, Jie He, Jeoffrey Auclair.

**Funding acquisition:** Jérôme Dupras.

**Project administration:** Jérôme Dupras.

**Supervision:** Jérôme Dupras.

**Validation:** Jérôme Dupras, Jie He.

**Writing – original draft:** Chloé L'Ecuyer-Sauvageau, Jeoffrey Auclair.

**Writing – review & editing:** Chloé L'Ecuyer-Sauvageau, Jérôme Dupras, Jie He, Charlène Kermagoret, Thomas G. Poder.

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
