## [Decision Letter · Decision Letter 0]

16 Jun 2020

PONE-D-20-10375

The economic value of the natural capital and ecosystem services of Canada's National Capital Green Network

PLOS ONE

Dear Dr. L'Ecuyer-Sauvageau,

Thank you for submitting your manuscript to PLOS ONE. After careful consideration, we feel that it has merit but does not fully meet PLOS ONE’s publication criteria as it currently stands. Therefore, we invite you to submit a revised version of the manuscript that addresses the points raised during the review process.

Please carefully consider all comments of reviewers and provide a clear response to each comment.

We look forward to receiving your revised manuscript.

Kind regards,

Neville Crossman, Ph.D.

Academic Editor

PLOS ONE

Journal Requirements:

2. Please amend either the title on the online submission form (via Edit Submission) or the title in the manuscript so that they are identical.

3. Please include captions for your Supporting Information files (as the Dupras et al. 2016.pdf file has been changed from "other" to "supporting information" item type) at the end of your manuscript, and update any in-text citations to match accordingly. Please see our Supporting Information guidelines for more information: http://journals.plos.org/plosone/s/supporting-information.

                      4. Please upload a copy of Figure 1, to which you refer in your text on page 17. If the figure is no longer to be included as part of the submission please remove all reference to it within the text.                                                  

Reviewers' comments:

Reviewer's Responses to Questions

**Comments to the Author**

1. Is the manuscript technically sound, and do the data support the conclusions?

Reviewer #1: Yes

Reviewer #2: Partly

2. Has the statistical analysis been performed appropriately and rigorously? 

Reviewer #1: Yes

Reviewer #2: No

3. Have the authors made all data underlying the findings in their manuscript fully available?

Reviewer #1: Yes

Reviewer #2: No

4. Is the manuscript presented in an intelligible fashion and written in standard English?

Reviewer #1: Yes

Reviewer #2: Yes

5. Review Comments to the Author

Reviewer #1: This paper presents an estimation of the value of ecosystem services of forests, wetlands, croplands, prairies and grasslands, and freshwater systems in Canada’s

Capital Region (Ottawa-Gatineau region).

Main comments:

This is an important study which I think will be highly cited. I admire the authors’ efforts to estimate so many ecosystem services in one study. It is important that such a study is done, otherwise we will continue to grossly undervalue the ecosystem services of natural assets. Ideally, the valuation of ecosystem services should be based on original studies. However, the authors’ have appropriately dealt with data, time, and budget constraints by applying a combination of methods instead of performing original studies. I believe the methods chosen by the authors which include market pricing, cost replacement, and two benefit transfer approaches (with adjustment and with meta-analysis) for ascertaining each of the ecosystem services of natural assets were appropriate ones.

I mostly have only minor comments for this paper - mainly for clarification. My only main comment for this paper would be for the authors to make a clear distinction - in the methods section - between the value of the stock of natural assets versus the value of the flow of the ecosystem services. This is consistent with the Systems of Environmental and Economic Accounting (SEEA) Framework (seea.un.org). For example, the stock value of trees would be based on the replacement cost of trees, which is different from the value of the flow of ecosystem services that trees provide e.g. carbon sequestration, storm water retention. If you don’t include the value of the stock of the asset, you will significantly under estimate its value.

I hope my comments will help improve the quality of this manuscript further.

Minor comments:

P6 Line 128: The description of the study site really needs a figure or a map to really help the readers who are not familiar with the area to get a good idea of what the land use composition looks like. Perhaps place the figure about here on P6.

P9 Line 203-205: It seems unusual that you have to make assumptions around population density to distinguish between urban and rural areas. I would have thought that Canada would have some sort of land zoning system that would describe the type of land use whether it is urban or rural.

P12 Line 260-261: How were the rates of dependence on pollinators calculated? Please describe the method, or provide a citation for the method.

P12 Line 262-265: I’m concerned that the recreational value is a gross under estimation of the true recreational value. And this value is being used to estimate the recreational value for both forests and wetlands. Would it be possible to capture visitor statistics and estimate the travel cost of visitors based on how far they have to travel to the site? This should only require the number of visitors per year and the post code of their residential address.

Table 1:

-For clarification, please specify what is the dependant variable (including units e.g. $/year, etc)?

-If I interpret the parameter value of [ln wetland size] correctly, its saying that the larger the wetland the lower the ecosystem services value. Please explain why this is the case for this study site. I would have expected a positive and/or quadratic parameter.

P20 Line 407: Please explain why the pollination value of urban and rural forests are assumed to be the same - at $31/ha/year. I would have expected them to be different.

P21 Line 428-429: Please provide citations for the ‘three monetary estimates’ of nutrient cycling.

P24 Line 494: It would be good if you could cite the studies where the values used in the benefit transfer method came from - unless they all came from He et al.

Table 9: Please add a column to specify the area (in ha) for each of the natural assets.

Figure 1: I was not able to see Figure 1 on the PDF copy that I have therefore I cannot comment.

Figure 2: You should put units in the legend of Figure 2.

Reviewer #2: The article start off by providing an excellent overview on the importance of ecosystem service (ES) in enabling decision makers to consider the impact of planning on people's well being. An extensive list of 13 Ecosystem services from the NCC's Greenbelt is provided. Several land uses and their spatial extent within the NCC Green network are provided. Four valuation methods are then given as the methods that will be used, however there is no explanation as to why these are chosen for the different ecosystem services.

After this there is a missing information between the land use covers and ecosystem services from these land use covers. The different land covers provide a lot more ecosystem services but there is no mention of why only a few were considered for valuation. For example wetlands have property price premium impact in urban areas, and this is not covered in this study.

A major issue with the paper is the limited information provided to support their estimated values. For example in the pollination benefit is just given as $31/ha/year but there is no explanation on what the counterfactual to this estimate is and there is no information on the crops/plants that benefit from this ES. For each reported benefit estimate, more information on the original study, attributes/indicators driving value and where possible a counterfactual scenario should be provided. For example, at the moment it is not clear whether the $31/ha/year pollination ES is additional yield gains or avoided costs of alternative pollination. This needs to be addressed for all reported benefits in the results section.

A table with land use covers, a list of ES by land use cover and valuation of methods will be a good improvement. This article has relied on previous work to inform their benefit estimation. There are many non-market valuation methods such as the choice modelling, hedonic pricing models, travel cost models which could provide robust and site specific benefits. However, these were not used and there is no explanation as to why this is the case.

A brief paragraph on limitations is required. Study limitations must cover why some ES are not quantified and why other more common methods e.g. travel cost for visits to forest were not used.

In summary, the article does not clearly explain their ES calculations approach, some equations will be helpful. There is also no acknowledgement or mention of the original studies in the results section where estimated values have been provide. More needs to be done to compare the study findings to what others have estimated for similar environmental assets and ES.

6. PLOS authors have the option to publish the peer review history of their article (what does this mean?). If published, this will include your full peer review and any attached files.

Reviewer #1: No

Reviewer #2: No

---

## [Author Response · Author response to Decision Letter 0]

31 Jul 2020

Response to Editors and Reviewers

Editors

Response: Thank you for sharing these templates, we used them to modify our submission. 

2. Please amend either the title on the online submission form (via Edit Submission) or the title in the manuscript so that they are identical.

Response: We will change this in this submission. 

3. Please include captions for your Supporting Information files (as the Dupras et al. 2016.pdf file has been changed from "other" to "supporting information" item type) at the end of your manuscript, and update any in-text citations to match accordingly. Please see our Supporting Information guidelines for more information: http://journals.plos.org/plosone/s/supporting-information.

Response: We added the citation for the Supporting information file.

4. Please upload a copy of Figure 1, to which you refer in your text on page 17. If the figure is no longer to be included as part of the submission please remove all reference to it within the text. 

Response: We added a copy of Figure 1 and used PACE to ensure it fits the journal’s requirements. 

Comments to the Author

1. Is the manuscript technically sound, and do the data support the conclusions?

Reviewer #1: Yes

Reviewer #2: Partly

2. Has the statistical analysis been performed appropriately and rigorously?

Reviewer #1: Yes

Reviewer #2: No

3. Have the authors made all data underlying the findings in their manuscript fully available?

Reviewer #1: Yes

Reviewer #2: No

4. Is the manuscript presented in an intelligible fashion and written in standard English?

Reviewer #1: Yes

Reviewer #2: Yes

5. Review Comments to the Author

Reviewer #1

Reviewer #1: This paper presents an estimation of the value of ecosystem services of forests, wetlands, croplands, prairies and grasslands, and freshwater systems in Canada’s

Capital Region (Ottawa-Gatineau region).

Main comments:

This is an important study which I think will be highly cited. I admire the authors’ efforts to estimate so many ecosystem services in one study. It is important that such a study is done, otherwise we will continue to grossly undervalue the ecosystem services of natural assets. Ideally, the valuation of ecosystem services should be based on original studies. However, the authors’ have appropriately dealt with data, time, and budget constraints by applying a combination of methods instead of performing original studies. I believe the methods chosen by the authors which include market pricing, cost replacement, and two benefit transfer approaches (with adjustment and with meta-analysis) for ascertaining each of the ecosystem services of natural assets were appropriate ones.

I mostly have only minor comments for this paper - mainly for clarification. My only main comment for this paper would be for the authors to make a clear distinction - in the methods section - between the value of the stock of natural assets versus the value of the flow of the ecosystem services. This is consistent with the Systems of Environmental and Economic Accounting (SEEA) Framework (seea.un.org). For example, the stock value of trees would be based on the replacement cost of trees, which is different from the value of the flow of ecosystem services that trees provide e.g. carbon sequestration, storm water retention. If you don’t include the value of the stock of the asset, you will significantly under estimate its value.

I hope my comments will help improve the quality of this manuscript further.

Response : Thank you for all of your comments and suggestions.

More specifically, to take into account the comment with regard to the distinction between stock and flow of resources, we added the following paragraph in section 3: 

“There is distinction to be made between the evaluation of stocks of natural resources and of flow of ecosystem services. Whereas the stock value of a wetland would be its replacement cost, the flow of the services provided by wetlands include flood prevention, carbon sequestration and habitat for biodiversity. In this study, we did include the values of the stock of wetlands, as the meta-analysis by He et al. [32] provides the value of the stock of one ha. When it comes to carbon storage, the value of the stock of carbon was taken into account for the service of Global climate regulation for Forests and for Wetlands. However, the value of the stock of capital was not included in the value of the other ES.”

We also added two sentences in our limitations paragraph in the Discussion: 

“Finally, our analysis was not exhaustive with regard to the valuation of the stock of natural capital, where only the values for Wetlands ES included the stock of natural capital and the values for the service of Global climate regulation included the value of the stock of carbon. This suggests that our estimate of the value of the NCC’s Green network is an underestimation.”

Minor comments:

1. P6 Line 128: The description of the study site really needs a figure or a map to really help the readers who are not familiar with the area to get a good idea of what the land use composition looks like. Perhaps place the figure about here on P6.

Response: Our Figure 1 shows the land use in the Green network, but to account for your comment, we added an element to show where this region is located in the Canadian landscape. We understand that this Figure was not properly uploaded at the time of our submission, so we will remediate to the situation at this time.

2. P9 Line 203-205: It seems unusual that you have to make assumptions around population density to distinguish between urban and rural areas. I would have thought that Canada would have some sort of land zoning system that would describe the type of land use whether it is urban or rural.

Response: Yes, Canada does use a land zoning system, however in our dataset the divisions are made at the census level, which often includes urban and rural areas. This is why we used Statistics Canada’s definition of urban and rural areas to define our cells. 

3. P12 Line 260-261: How were the rates of dependence on pollinators calculated? Please describe the method, or provide a citation for the method.

Response: The rates of dependence on pollinators were not calculated; we took the rates provided by Morse and Calderone (2000) as reported by AAC (2017). The method for the estimation of the pollination service is being explained in more detail in the Market pricing subsection. 

“The economic valuation of the pollination service was based on the method described in Winfree et al. [53], where the value of the pollinating service from wild pollinators, in this case, is based on the value of production. As our focus was on wild pollinators, we used the hypothesis from the study by Rands and Whitney [54] which studies the effect of wild field hedges for nesting pollinators and their travel capacity which varies from 125 m to 2 km. As a conservative distance, we chose to consider the distance of 1 km that pollinators can travel towards agricultural lands from nesting sites. By doing so, we estimated the pollinating potential of wild field hedges (forests and prairies and grassland) on the croplands. This was done by counting the area in a 1 km buffer around the crops dependent on pollination that contains is made of forests and prairie or grassland (surfaceFPG). The equation we used is the following: 

Vpollination = ((P * Y – C)* D *ρ) * surfaceFPG (1)

Where Vpollination is the value of the pollination service, P is the price of the crop, Y is the yield, C are the costs of production, D is the crop dependency on insect pollinators, and ρ is the percent reduction in insect pollinators [53]. In our case, we assumed that the value of ρ is 1. We limited our estimation to crops that have a dependence on pollinators. We used the value of the net benefit (P*Y – C) from beans, berries, soy, and forage as calculated for the provisioning service, and took into account their surface area and their rates of dependence on pollinators. We used the degree of dependence as reported by Morse and Calderone [55] [56] for berries/strawberries (0.2), soy (0.1), and forage (1). For beans, although the degree of dependence was not provided in Morse and Calderone [55], we assumed a degree of dependence of 0.1, as it is a flowering crop.”

4. P12 Line 262-265: I’m concerned that the recreational value is a gross under estimation of the true recreational value. And this value is being used to estimate the recreational value for both forests and wetlands. Would it be possible to capture visitor statistics and estimate the travel cost of visitors based on how far they have to travel to the site? This should only require the number of visitors per year and the post code of their residential address.

Response: You are very right to point out the fact that our recreational value is a gross underestimation. We would have liked to do a travel cost analysis, but we do not have access to the postal codes of visitors. At the Gatineau Park and in other green infrastructures of the Green network, access is usually free and few people purchase passes, mainly to access the park in the winter. However, there was a study conducted in 2015-2016 by Environics, a firm specialized in surveys. They did not perform a true travel cost analysis, but they did report the amount of money spent by visitors of the Gatineau Park in the region ($184M). As a result, we will change our value for the recreation ES to reflect this spending instead of simply using the entrance fees ($2.7M). We recognize that this value may still be an underrepresentation of the true recreational potential of the Green Network as it only focuses on the Gatineau Park, but it is more interesting than our previous estimate.

5. Table 1: 

-For clarification, please specify what is the dependant variable (including units e.g. $/year, etc)?

Response: The value of the dependent variable is the natural logarithm of the value of the stock of 1 ha of wetland ($/ha). 

6. If I interpret the parameter value of [ln wetland size] correctly, its saying that the larger the wetland the lower the ecosystem services value. Please explain why this is the case for this study site. I would have expected a positive and/or quadratic parameter.

Response: In He et al.’s (2015) paper, they explain that the wetland size has a decreasing marginal utility, where the larger the wetland, the smaller the new value of an additional hectare of wetland. In this sense, the value of [ln wetland size] means that as the size of a wetland increases, the ecosystem services value does not increase in the same manner. 

7. P20 Line 407: Please explain why the pollination value of urban and rural forests are assumed to be the same - at $31/ha/year. I would have expected them to be different.

Response: The value of the pollination service was estimated by dividing the value of production of crops that are dependent on pollination by the area in the 1 km buffer zone that is forested or a prairie or pasture. This hypothesis was based on idea that wild pollinators can establish their nesting area in these land uses and then travel 1 km (125m to 2 km – Rands and Whitney 2011) to croplands. In this sense, we did not distinguish between land uses. 

8. P21 Line 428-429: Please provide citations for the ‘three monetary estimates’ of nutrient cycling.

Response: As you will see in the reviewed version of the manuscript, we have added all of the sources that were used for the benefit transfer analysis with adjustment in Table 2 and in S.1 Table. 

9. P24 Line 494: It would be good if you could cite the studies where the values used in the benefit transfer method came from - unless they all came from He et al.

Response: As you will see in the reviewed version of the manuscript, we have added all of the sources that were used for the benefit transfer analysis with adjustment in Table 2 and in S.1 Table. 

10. Table 9: Please add a column to specify the area (in ha) for each of the natural assets.

Response: We added a row, after the total, to specify the area (in ha) for each of the natural assets.

11. Figure 1: I was not able to see Figure 1 on the PDF copy that I have therefore I cannot comment. 

Response: Sorry about this situation, it will be corrected in this revision.

12. Figure 2: You should put units in the legend of Figure 2.

Response: Thank you for your comment, we added the units in the legend title of Figure 2. 

Reviewer #2

Reviewer #2: The article start off by providing an excellent overview on the importance of ecosystem service (ES) in enabling decision makers to consider the impact of planning on people's well being. An extensive list of 13 Ecosystem services from the NCC's Greenbelt is provided. Several land uses and their spatial extent within the NCC Green network are provided. Four valuation methods are then given as the methods that will be used, however there is no explanation as to why these are chosen for the different ecosystem services.

After this there is a missing information between the land use covers and ecosystem services from these land use covers. The different land covers provide a lot more ecosystem services but there is no mention of why only a few were considered for valuation. For example wetlands have property price premium impact in urban areas, and this is not covered in this study.

A major issue with the paper is the limited information provided to support their estimated values. For example in the pollination benefit is just given as $31/ha/year but there is no explanation on what the counterfactual to this estimate is and there is no information on the crops/plants that benefit from this ES. For each reported benefit estimate, more information on the original study, attributes/indicators driving value and where possible a counterfactual scenario should be provided. For example, at the moment it is not clear whether the $31/ha/year pollination ES is additional yield gains or avoided costs of alternative pollination. This needs to be addressed for all reported benefits in the results section.

A table with land use covers, a list of ES by land use cover and valuation of methods will be a good improvement. This article has relied on previous work to inform their benefit estimation. There are many non-market valuation methods such as the choice modelling, hedonic pricing models, travel cost models which could provide robust and site specific benefits. However, these were not used and there is no explanation as to why this is the case.

A brief paragraph on limitations is required. Study limitations must cover why some ES are not quantified and why other more common methods e.g. travel cost for visits to forest were not used.

In summary, the article does not clearly explain their ES calculations approach, some equations will be helpful. There is also no acknowledgement or mention of the original studies in the results section where estimated values have been provide. More needs to be done to compare the study findings to what others have estimated for similar environmental assets and ES.

Response to general comments: 

Firstly, we would like to thank you for all of your comments. 

We have responded more specifically to your comment about the relationship between land cover and the methods used in your specific comment #6, and to your comment about the importance of naming our sources in your comment #11.

About our choice of methods, we realize that we could have been clearer. Our choice of methods was based on time and resources constraints. To calculate such an important number of ES in one study and in a short period, we needed to use approaches that were not too resource intensive. Also, we wanted to provide the NCC with values that showcases the importance of ES to the Ottawa-Gatineau region, as part of a case to keep protecting these natural areas, and not to provide specific values on each ES. This was more an exercise designed to raise awareness. To clarify this in the text, we added a sentence in the Materials and methods section: 

“This choice of methods was based in part on time and resource constraints, and in part on the purpose of this study, which was to raise awareness on the general importance of ES in the NCC’s Green network.”

A brief paragraph on limitations was already included in the discussion section (the second to last paragraph). 

“These results should nevertheless be considered in light of methodological limitations inherent to the approach chosen in this study, especially with regard to the spatial analysis and the economic valuation. Data availability was a limiting factor in both the spatial analysis and the economic valuation. Although tests were made to detect errors in the GIS layer, the chosen source only had an 85% precision. Data availability also impacted the economic valuation, especially when using the benefit transfer approach. Even though care was taken to avoid transfer bias, the results are not as robust as a study based exclusively on primary data. Additionally, the benefit transfer analysis was dependent on published literature, for which the coverage of ecosystems and ES is unequal. This unequal attention affects the accuracy of some monetary estimates. For example, there were more studies on forests and wetlands than on freshwater and agricultural lands. [Finally, our analysis was not exhaustive with regard to the valuation of the stock of natural capital, where only the values for Wetlands ES included the stock of natural capital and the values for the service of Global climate regulation included the value of the stock of carbon. This suggests that our estimate of the value of the NCC’s Green network is an underestimation.] ”

Despite the presence of this paragraph, we do recognize that a greater acknowledgement of the limits of our analysis can be made clearer in the text. We believe that such an acknowledgement will be made clearer with Table 1 that replace the list of ES valued. 

Specific comments (reviewer 2)

1. Line 161 – worth noting that raising taxes is relatively difficult and requires political support.

Response: Yes, this is a good point. In order to nuance the initial statement, we have added the following sentence on 

Lines 164-165: 

“Indeed, it is often politically damaging for municipal governments to raise new taxes or to increase the taxation level of citizens.”

2. Line 168 – 172 - Apart from having a lower risk of being developed, are there any other attributes to the NCC that make it to have potentially different end-user values to the open spaces managed by municipalities? Otherwise, the economic value should not be different just because of who manages.

Response: Yes, the economic value should not be different because the lands are managed by the NCC, but this study was started in 2015 when the Canadian Premier was Stephen Harper. At this time, the budget allocated to the NCC was less important and it seemed a good idea to present the case for maintaining the current land area of the Green Network through an economic values lens. 

To reflect this issue of political pressure, we have adapted the text on Lines 165 – 171: 

“The case of the NCC’s green network is singular in the sense that the largest green network in the Ottawa-Gatineau region is not managed by the municipalities, but by an agency of the federal government that is not influenced by the same pressures. It can, however, be indirectly influenced by political pressures, through budget allocation and changes to its mandate. In this case, assessing the economic value of the lands preserved by the NCC, can help to showcase the importance of natural capital to human well-being in the region.”

3. Line 175 – “cost” should be “replacement cost”?

Response: Yes, thank you for noticing this. We changed “cost” to “replacement cost”.

4. Line 176 – delete “of”

Response: ok. 

5. Line 218 – This is the first mention of your database, explain it here or let the reader know that this is described later

Response: As suggested, we have added a description of the database at its first mention. 

“Our selection was also based on our own database. This database was first developed as part of a literature review by Dupras [35] which included studies of ecosystem services valuation on forest, wetland and agricultural ecosystems. In the context of this study, the database was supplemented with a literature review of the economic valuation exercises of ES that had been undertaken in Quebec and Ontario [e.g. 9, 18, 36, 37] and with studies on the valuation of aquatic ecosystems. The selected studies from this database are presented in S1 Table in the Supporting information.”

6. Line 218 – you mention that 13 ES were selected. Explain the basis for this selection and provide information on ES that were dropped.

Response: The selection of the ES was based on whether or not our database included at least one estimate for the value for the given ecosystem. In some cases, we were able to include additional ES, because of data availability, as in the case of agricultural products, for example. 

To clarify our rationale for only discussing these ES, we added a justification in the text.

“13 ES were selected for further analysis in the context of the NCC’s Green Network. Although the ecosystems under study generate more ecosystem services, we limited our analysis to the ecosystem services whose value had been estimated previously and to ecosystem services for which we could estimate the value using the data available in the allotted time.”

To outline the caveat that there are ES generated by the ecosystems that are not presented in the study, we changed our presentation of the list of ES on Lines 235-252. In the new table, where no data was available, we added a hollow dot. 

Table 1. Selection of ES based on ecosystems and data available.

Ecosystem services Ecosystems / Land uses

 Urban forest Rural forest Wetlands Cropland Prairie and grassland Freshwater

Provisioning services

Agricultural products ● ● 

Other food ○ ○ ○ ○

Raw material ○ ○ ○ ○ ○ ○

Genetic diversity ○ ○ ○ ○ ○ ○

Regulating services

Global climate regulation ● ● ● ○ ● ○

Air quality ● ● ○ ○ ○ ○

Water provisioning ● ● ● ○ ○ ○

Waste treatment ● ● ● ○ ○ ●

Erosion control ● ● ● ● ○

Pollination ● ● ○ ○ ● ○

Habitat for biodiversity ● ● ● ○ ● ●

Disturbance / natural hazards prevention ● ● ● ○ ○ ○

Pest management ● ● ○ ○ ○ ○

Nutrient cycle ● ● ○ ● ● ○

Cultural services

Aesthetics (landscape) ○ ○ ○ ● ● ○

Recreational activities and tourism ● ● ● ● ● ●

Scientific and educational ○ ○ ○ ○ ○ ○

Spiritual ○ ○ ○ ○ ○ ○

Heritage and cultural identity ○ ○ ○ ○ ○ ○

7. Line 237 – 242 – This introduction requires adjustment to re-focus on valuation methods and not just those that were used in the study. Thus, an overview of non-market valuation techniques followed by a discussion of the one used in the study plus an explanation of why they were chosen should be provided.

Response: Thank you for your comment. To explain why we chose these methods over others, we moved a sentence from section 3 to section 3.3. 

We also added a paragraph on section 3.3 to provide more information on non-market valuation methods, as you suggested. In any case, this inclusion is useful, as some of the studies that we use to obtain the values as part of the benefit transfer method (using the database) also use stated and revealed preference methods. Since we will add more detail about the methods used in studies that we rely upon to obtain the values per ecosystem service, it will be helpful to the reader. 

“The analysis of the 13 ES mentioned above was carried out using four methodological approaches. The market pricing method was used to determine the economic value of agricultural products, pollination, and recreational activities and tourism. For all of the other non-market ES, the benefit transfer approach with adjustment, the benefit transfer method with meta-analysis, and the replacement cost method were used. [Considering the large number of ecosystems that compose the NCC’s green network and the quantity of ES that they produced, the use of these methods enabled us to perform the analysis given resource constraints (time, especially), and carry out the main objective of this study. These methods will be discussed in the subsections below. ] taken from section 3.

In the case where we would have had time to calculate a value that is more specific to our study area, we could have used other methods. Indirect valuation methods include stated and revealed preference approaches. Revealed preference approaches are based on observed behavior [38]. They include the travel cost method, where the value of recreation, for example, is estimated as a function of the distance traveled to a location and of the purchases made on the way to and at the location [39]. They also include the hedonic pricing method, where the aesthetics ecosystem service, for example, can be estimated based on the premium associated to the value of a house that is located in a specific landscape [39]. The stated preference approaches are based on stated behavior, generally gathered as part of a survey, where hypothetical scenarios are presented to individuals. They include contingent valuation and choice experiment methods, where respondents are asked, for example, how much they are willing to pay to preserve an aesthetically pleasing landscape (contingent valuation) or are asked to choose their favorite option of a recreational area, where each scenario is composed of a number of given attributes (choice experiment) [40]. For more information on the methods and suggestions about the most appropriate ones to use in a specific context, we suggest consulting the Ecosystem Services Toolkit [41] or the TESSA [42].”

8. Line 254 – 261 what is the counterfactual for this estimate. What production crops are affected and what is the benefit being measured? Is it avoided costs of alternative pollination methods or increases in yield? These questions should be addressed. A similar approach needs to be adopted for all other benefits/ES.

Response: We used the value of production from wild pollination to estimate the value of this service. To account for your comment and provide information about the way this service was calculated, we changed the paragraph starting on Line 254 to the text below. 

As for the other services, we also added more information about the methods used in the primary studies when the benefit transfer method was used in a table in the Appendix as well as in Table 2. We also added a discussion on how it compares to other values when the values do not come from benefit transfer. 

“The economic valuation of the pollination service was based on the method described in Winfree et al. (2011), where the value of the pollinating service from wild pollinators, in this case, is based on the value of production. As our focus was on wild pollinators, we used the hypothesis from the study by Rands and Whitney [47] which studies the effect of wild field hedges for nesting pollinators and their travel capacity which varies from 125 m to 2 km. As a conservative distance, we chose to consider the distance of 1 km that pollinators can travel towards agricultural lands from nesting sites. By doing so, we estimated the pollinating potential of wild field hedges (forests and prairies and grassland) on the croplands. This was done by counting the area in a 1 km buffer around the crops dependent on pollination that contains is made of forests and prairie or grassland (surfaceFPG). The equation we used is the following: 

Vpollination = ((P * Y – C)* D *ρ) * surfaceFPG

Where Vpollination is the value of the pollination service, P is the price of the crop, Y is the yield, C are the costs of production, D is the crop dependency on insect pollinators, and ρ is the percent reduction in insect pollinators (Winfree et al. 2011). In our case, we assumed that the value of ρ is 1. We limited our estimation to crops that have a dependence on pollinators. We used the value of the net benefit (P*Y – C) from beans, berries, soy, and forage as calculated for the provisioning service, and took into account their surface area and their rates of dependence on pollinators. We used the degree of dependence as reported by Morse and Calderone (2000, from AAC 2017) for berries/strawberries (0.2), soy (0.1), and forage (1). For beans, although the degree of dependence was not provided in AAC (2017), we assumed a degree of dependence of 0.1, as it is a flowering crop.”

9. Line 296 – Clarify what is meant by adjustment here. Is it the conversion of foreign currency Canadian dollars or adjusting dollar values for inflation? This should be made clear.

Response: To clarify what we meant by adjustment, we added the following sentence after Line 296: 

“When the values were in a currency other than Canadian dollars, we performed a first adjustment using Purchasing Power Parity (OECD), then we adjusted all values in Canadian dollars using the inflation rate (Bank of Canada).”

10. Line 367 (Table 2) – since the paper is about valuing ES. A table with information on Land use cover, ecosystem services and valuation will be more informative for the reader.

Response: To make Table 2 more informative, we modified it. It now provides the list of ES per Land use cover with the valuation method, and the number of ha per Land Use. When we used the benefit transfer with adjustment method, we also added the number of values used, the sources and the methods used in the primary studies. 

11. Results section – all your ES estimates need a clear explanation of what the estimated values are. For example, Line 421 what exactly does disturbance or flood control mean, what parameters were used to arrive at $5030/ha/year, given that this is a benefit transfer exercise what is the original source of your estimate. This source must be acknowledged as a citation. This comment applies to results in the article. 

Response: Thank you for your comment. To better acknowledge all of our sources, we added references to primary sources in the text, as well as in Table 2 and in Table S.1. We also added more information about the meaning of different ecosystem in the text. 

12. Results section – for each of the ES quantified, how do your findings compare to previous studies?

Response: Thank you for your suggestion. We decided to compare our results for values that did not come from the benefit transfer with adjustment method, as these already reflect the values generated from other studies. We added these comparison in the results section. 

13. Line 596 – 598 – this is an important point. However, the study has not sufficiently supported the methods they used or provided enough information for their methods to be assessed in a meaningful way. This makes it difficult to accept that the study has satisfactorily contributed to the literate as they articulate in Line 596 – 598 (The originality of this study is to explore the value of a suite of regulating services while in many studies, these services are often left aside because of their difficult association with economic indicators.)

Response: In line with your comment, we changed the formulation of these line to reflect the fact that we aimed to achieved this and that regulating services are important to take into account in ES valuation studies. 

“The aim of this study was in part to explore the value of a suite of regulating services, since in many studies, these services are often left aside because of their difficult association with economic indicators.”

14. Figure 1: What do the different colors in this figure mean?

Response: The colors refer to land uses. We added a legend to this figure so that it becomes clear to the reader. 

General comments:

When we first performed the benefit transfer analysis with adjustment, we did not use the right reference year for values that came from the study by Dupras et Alam (2014). Although the values were presented in Canadian dollars from 2010, we used the year 2013 as a starting point to convert the values in 2015 $ CAD. As such, the pest management/biological control service urban forests and for prairie and grasslands were changed from 42$/ha/year to 45$/ha/year. For croplands, the erosion control service was changed from 106$/ha/year to 109$/ha/year, the nutrient cycling service changed from 174 to 184$/ha/year, and the recreation service changed from 88 to 94$/ha/year. Finally, for prairie and grasslands, we also adjusted the ES of habitat for biodiversity from 2324 to 2467$/ha/year. This change has an impact on the total economic value, but the biggest change to this value comes from the fact that we used a new data to estimate the recreational value, in line with Reviewer 1’s suggestion. This value was not initially available, since the publication of the report occurred after we were done with this initial study. However, it better represents the recreational value of ecosystems in the NCC Green network, which is why we believe it is interesting to change our initial value estimate to this one. In addition, some of the initial values have been changed when using the benefit transfer with adjustment method due to the rounding of values.

---

## [Decision Letter · Decision Letter 1]

5 Oct 2020

PONE-D-20-10375R1

The economic value of Canada’s National Capital Green Network

PLOS ONE

Dear Dr. L'Ecuyer-Sauvageau,

Thank you for submitting your manuscript to PLOS ONE. After careful consideration, we feel that it has merit but does not fully meet PLOS ONE’s publication criteria as it currently stands. Therefore, we invite you to submit a revised version of the manuscript that addresses the points raised during the review process.

We look forward to receiving your revised manuscript.

Kind regards,

Neville Crossman, Ph.D.

Academic Editor

PLOS ONE

Additional Editor Comments (if provided):

While Reviewer #2 has provided further numerous comments, they are all relatively minor and should not require significant work.

Reviewers' comments:

Reviewer's Responses to Questions

**Comments to the Author**

1. If the authors have adequately addressed your comments raised in a previous round of review and you feel that this manuscript is now acceptable for publication, you may indicate that here to bypass the “Comments to the Author” section, enter your conflict of interest statement in the “Confidential to Editor” section, and submit your "Accept" recommendation.

Reviewer #1: (No Response)

Reviewer #2: (No Response)

2. Is the manuscript technically sound, and do the data support the conclusions?

Reviewer #1: Yes

Reviewer #2: Partly

3. Has the statistical analysis been performed appropriately and rigorously? 

Reviewer #1: Yes

Reviewer #2: No

4. Have the authors made all data underlying the findings in their manuscript fully available?

Reviewer #1: Yes

Reviewer #2: No

5. Is the manuscript presented in an intelligible fashion and written in standard English?

Reviewer #1: Yes

Reviewer #2: Yes

6. Review Comments to the Author

Reviewer #1: Additional comments

The authors have made substantial improvements to the manuscript. I just have a few more questions and requests that I would like the authors to respond to in order to clarify their methods and assumptions further.

1. Is it possible to draw (e.g. using dotted lines) where Gatineau Park and the Greenbelt are in Figure 1?

2. Is Figure 2 mislabelled? I do not see any land use that is in gray.

3. The authors said that “ρ is the percent reduction in insect pollinators [53]. In our case, we assumed that the value of ρ is 1”. If you assumed in your formula Vpollination = ((P * Y – C)* D *ρ) * surfaceFPG that ρ=1, I just wanted to confirm whether ρ is equal to 1%, or is ρ equal to 100%”.

4. In your response to my question regarding the dependant variable of the explanatory variables described in Table 2 - you said the dependant variable is “the natural logarithm of the value of the stock of 1 ha of wetland ($/ha)”. However, in your manuscript, above Table 2, you wrote “The explanatory variables (value of the natural logarithm of the stock of natural capital”. I just wanted to confirm that this is correct because it means that you have an equation that has a natural log on both the LHS (i.e. the dependant variable) and the RHS (i.e. the explanatory variables) of the equation.

5. I think its best if the authors formally explain the differences between the stock and flow of ES. The definition provided by Jones et al. (2016) is clear and accurate and perhaps you can use a definition along these lines. Jones et al. said “Natural capital has been variously defined as the stock of physical assets in the environment (water, trees, minerals, species, etc.), but also the processes (e.g. water purification, climate regulation) from which we obtain benefits (e.g. NCC, 2013).”

References

L. Jones, L. Norton, Z. Austin, A.L. Browne, D. Donovan, B.A. Emmett, Z.J Grabowski, D.C. Howard, J.P.G. Jones, J.O Kenter, W. Manley, C. Morris, D.A. Robinson, C. Short, G.M. Siriwardena, C.J. Stevens, J. Storkey, R.D. Waters, G.F. Willis. (2016). Stocks and flows of natural and human-derived capital in ecosystem services, Land Use Policy, 52 (March 2016): 151-162, https://doi.org/10.1016/j.landusepol.2015.12.014.

Reviewer #2: This authors are attempting to publish a potentially very important paper on the non-market valuation of environmental assets. They do cover a broad range of ecosystem services. Unfortunately, some of the approaches used are either not well explained or potentially not appropriate.

For this paper to have a positive impact to readers, practitioners and/or researchers the authors should ensure that all their quantification approaches are suited to the benefits they are attempting to value. Where possible, equations and descriptions for all variables used in the estimation process should be well documented. Another important point is that at times the paper is written as if it must be read in conjunction with multiple previous studies. However, once the estimation approaches and variables are explained, this will not be a problem as the paper will be detailed and informative enough.

Below are my specific comments on the new version of the paper.

Page 4 of 147 point #4

The proposed method here is a cause of concern. While the money is spent by park visitors this money is not a reflection of the economic value of the park. The economic value of the park should be the main focus if we are seeking to put monetary values to ecosystem service. In the absence of the right data to perform a travel cost model type of analysis, a benefit transfer approach from a similar study will be your next best option for valuing the recreation ecosystem service/benefit. By using the money spent by visitors in the region, you are potentially overstating the economic value because (1) you are using gross expenditure values, (2) do not account for or attribute any of the expenditure to other nearby attractions/activities that people engage in and pay for but these are not necessarily park-based. I would suggest that you do a consumer surplus benefit transfer approach for recreation.

P8 of 147 point #8

Your pollination ES quantification approach implies that without the park the nearby agricultural activities will not exist, that is there will be no production? Is this the right counterfactual for you case study areas, is it possible that agricultural production could still exist will lower native pollinators? In short, the agricultural value of production is potential erroneous without a clear and relevant counterfactual.

P28 of 147 Line 271

This approach provides the gross estimate. However, the correct values should be the marginal e.g. without this park nearby farmers may have to hire bees for pollination, they may have lower yields. In either case, you need to properly frame your counter-factual. Or is the counter-factual that these croplands will not produce anything without the park being located within 2km. It should be straight forward to account for high production costs of lower yields in your equation in line 281

P28 of 147 Line 278

Consider replacing counting be measuring?

P28 of 147 Line 282

The assumption that urban and rural forests have the same pollination contribution is a bit of a weakness. One would have thought that rural forests will be a richer habitat for native bees than urban forests and thus rural forests will have higher populations of bee/insect pollinators?

Could rho (line 281) be used to account for lower native bee pollinators when the park is not there or when it is there but of lower quality compared to its current status.

Also, could rho be used to differentiate between urban and rural forests?

The authors do allude to this fact in line 407-408.

P29 of 147 Line 294 - 296

Please explain why the Dupras et al values are most suited for your recreation estimates. A brief explanation here will be helpful for the reader.

P29 of 147 Line 297 to 306

This section only covers the social cost of carbon, how were the biophysical parameters estimated (i.e. tonnes of carbon stored and sequestered)?

P29 of 147 Line 297 to 305

What was the appraisal period for your discounted cash-flow?

P37 of 147 Line 437 to 444

Why was this approach adopted? Both the quality of the environment asset and production value of crops affect the magnitude of the estimated benefit through pollination. Both are important in putting a dollar value on the pollination ecosystem service.

P38 of 147 Line 464 “calculating” should this be calculated?

P38 of 147 Line 471

Please name and describe the “three monetary estimates” here.

P38 of 147 Line 476

Given that these are benefit transfer values, please explain what they represent within your paper as well as having their source. Even in a benefit transfer paper, it is still very important for your readers to have an idea of what a $318/ha/year for waste treatment means without the need to look at other papers. This comment applies to all your benefit transfer estimates in the paper.

P40 of 147 Line 500

Can you please include an equation showing all variables used for this estimation. It will also be helpful to show the reader your input values for the calculations.

P40 of 147 Line 510-511

While you provide a citation. Please explain the basis for the value here.

P40 of 147 Line 527

I am unclear on what approach was used for the Gatineau Park analysis to yield the $3,338/ha/yr.

Also, the $19/ha/year to $10,741/ha/year is a very wide range. Do both numbers represent the willingness to for pay recreation at a park similar to your study area?

P42 of 147 Line 548

Is this a gross or net value of production?

P42 of 147 Line 549

Please explain the adjustment entails.

P43 of 147 Line 555

Should the total in the first row and total value column be 4.592 rather than 4592.0?

P46 of 147 Line 621

Are the dollar values in a per ha basis

P46 of 147 Line 624

What is meant by K$ if thousands please specify fully, K$ often used to refer to constant dollars.

P48 of 147 Line 624

Should this line start with “66% of the estimated economic value ...”

7. PLOS authors have the option to publish the peer review history of their article (what does this mean?). If published, this will include your full peer review and any attached files.

Reviewer #1: No

Reviewer #2: No

---

## [Author Response · Author response to Decision Letter 1]

20 Nov 2020

Reviewer #1: Additional comments

The authors have made substantial improvements to the manuscript. I just have a few more questions and requests that I would like the authors to respond to in order to clarify their methods and assumptions further.

1. Is it possible to draw (e.g. using dotted lines) where Gatineau Park and the Greenbelt are in Figure 1?

Answer: Unfortunately, we do not have the exact limits of the Gatineau Park, and this information is not available on the web in a format that we can use in ArcGIS. In that case, although we have the limits of the Greenbelt, it may be preferable not to include this information in the map.

2. Is Figure 2 mislabelled? I do not see any land use that is in gray.

Answer: No, the features in gray in Figure 2 represent land that we did not include in our analysis, for example land outside of the study area and urban lands. We included a caption in the legend to specify this in Figure 2. 

3. The authors said that “ρ is the percent reduction in insect pollinators [53]. In our case, we assumed that the value of ρ is 1”. If you assumed in your formula Vpollination = ((P * Y – C)* D *ρ) * surfaceFPG that ρ=1, I just wanted to confirm whether ρ is equal to 1%, or is ρ equal to 100%”.

Answer: Thank you for mentioning this. We initially used a value of 100% for ρ, but in the formula this is expressed in decimals. However, considering the comments and suggestions of Reviewer 2 (#2 and 5), we have changed this value. There is now a rho value for specific crop types and we used the inverse of the value provided by Morse and Calderone [55]. We have also used half of these values in urban areas, to reflect the decrease in the number of native pollinators, following the results by Fetridge et al. (2008).

4. In your response to my question regarding the dependant variable of the explanatory variables described in Table 2 - you said the dependant variable is “the natural logarithm of the value of the stock of 1 ha of wetland ($/ha)”. However, in your manuscript, above Table 2, you wrote “The explanatory variables (value of the natural logarithm of the stock of natural capital”. I just wanted to confirm that this is correct because it means that you have an equation that has a natural log on both the LHS (i.e. the dependant variable) and the RHS (i.e. the explanatory variables) of the equation.

Answer: You are right, this is a mistake. Thank you for noticing. This is the text that should have been written in the first place. We also added the equation that is used to perform the analysis. 

“The explanatory variables and their associated coefficients used as part of this meta-analysis are presented in Table 2. Equation (1) was used to calculate values of the ES. The values of the constant â and of the coefficients b ^var had been estimated by the meta-analysis [30].

Y ^j=exp⁡(a ^ )*exp⁡(b ^servXservj)*exp⁡(b ^wXwj)*exp⁡(b ^geoXgeoj)*exp⁡(b ^ecoXecoj)*exp⁡(b ^typeXtypej) (1)

In the equation, j refers to each wetland that was assessed. The dependent variable (Ŷ) is represented by the value of the natural logarithm of the stock of natural capital of 1 ha of wetland.”

5. I think its best if the authors formally explain the differences between the stock and flow of ES. The definition provided by Jones et al. (2016) is clear and accurate and perhaps you can use a definition along these lines. Jones et al. said “Natural capital has been variously defined as the stock of physical assets in the environment (water, trees, minerals, species, etc.), but also the processes (e.g. water purification, climate regulation) from which we obtain benefits (e.g. NCC, 2013).”

References

L. Jones, L. Norton, Z. Austin, A.L. Browne, D. Donovan, B.A. Emmett, Z.J Grabowski, D.C. Howard, J.P.G. Jones, J.O Kenter, W. Manley, C. Morris, D.A. Robinson, C. Short, G.M. Siriwardena, C.J. Stevens, J. Storkey, R.D. Waters, G.F. Willis. (2016). Stocks and flows of natural and human-derived capital in ecosystem services, Land Use Policy, 52 (March 2016): 151-162, https://doi.org/10.1016/j.landusepol.2015.12.014.

Answer: Thank you for this suggestion. We adjusted the text to attempt to make this distinction clearer. 

“There is a distinction to be made between the evaluation of stocks of natural resources and of flow of ES. For Jones et al. [2016], natural capital encompasses the ‘stocks’, which represent assets found in the environment, and the processes through which humans perceive benefits, which can be regarded as ‘flows’ or as transformations or evolutions of the stocks [Jones et al. 2016: 154].”

Additional information for Reviewer 1: We went back with our initial estimate of the recreational ecosystem service. As pointed out by Reviewer 2 in its comment #1, using the value provided by the Environics report does not provide a marginal value for the recreational service. As a result, we decided to go back to our initial estimate, based only on access fees, even though we recognize it can only provide a partial value. Nevertheless, this value can be directly tied to recreational activities carried out in the Gatineau Park, which is why we believe it is more suitable in the context of this study. 

Reviewer #2: This authors are attempting to publish a potentially very important paper on the non-market valuation of environmental assets. They do cover a broad range of ecosystem services. Unfortunately, some of the approaches used are either not well explained or potentially not appropriate.

For this paper to have a positive impact to readers, practitioners and/or researchers the authors should ensure that all their quantification approaches are suited to the benefits they are attempting to value. Where possible, equations and descriptions for all variables used in the estimation process should be well documented. Another important point is that at times the paper is written as if it must be read in conjunction with multiple previous studies. However, once the estimation approaches and variables are explained, this will not be a problem as the paper will be detailed and informative enough.

Below are my specific comments on the new version of the paper.

1. Page 4 of 147 point #4

The proposed method here is a cause of concern. While the money is spent by park visitors this money is not a reflection of the economic value of the park. The economic value of the park should be the main focus if we are seeking to put monetary values to ecosystem service. In the absence of the right data to perform a travel cost model type of analysis, a benefit transfer approach from a similar study will be your next best option for valuing the recreation ecosystem service/benefit. By using the money spent by visitors in the region, you are potentially overstating the economic value because (1) you are using gross expenditure values, (2) do not account for or attribute any of the expenditure to other nearby attractions/activities that people engage in and pay for but these are not necessarily park-based. I would suggest that you do a consumer surplus benefit transfer approach for recreation.

Answer: Taking your comment into account, we have decided to go back to our previous estimate of the value of the recreation service, as it relied only on money spent to perform some recreational activities in the Park. 

“The economic value associated to recreational activities and tourism was estimated based on results from the NCC’s 2014-2015 Annual Report [89] which showed that the NCC collected $2.7 million in user fees, which is equivalent to $75/ha/year when the $2.7 million value is divided over the Gatineau Park’s entire spatial area (36,161 ha). These user fees correspond to annual passes used to access the park during the winter to perform cross-country skiing and snowshoeing, to camping fees and to parking fees within specific areas of the park. When we compare this value to a report by Environics [27], which showed that Gatineau park visitors (77% of residents and 23% of non-residents) spent $184 million in the region as part of their travel to the Gatineau Park in food, sport supply and other purchases such as gas, the $2.7 million in user fees seems quite low. As a result, the value of $75/ha/year is most likely insufficient to represent the real value of the recreational potential of the Gatineau Park. It is nevertheless the only value that we can directly tie to activities performed in the Park.”

2. P8 of 147 point #8

Your pollination ES quantification approach implies that without the park the nearby agricultural activities will not exist, that is there will be no production? Is this the right counterfactual for you case study areas, is it possible that agricultural production could still exist will lower native pollinators? In short, the agricultural value of production is potential erroneous without a clear and relevant counterfactual.

Answer: Using the production value method, the value of pollination is estimated with the assumption that there is no substitute pollinators, from honey bees or other pollinators in this case. However, it does not mean that there will be no production, for most crops, because we take into account the level of crop dependency on crop pollination (D). This level of dependency determines to what extent crops need pollinators. In the case of forage, the degree of dependency on pollinators is set at 1 (fully dependent on pollinators) (Morse and Calderone 55), which would mean that without pollinators, there would be no production. As for the other crops that depend on pollination, their level of dependence is less: 0.2 for strawberries, 0.1 for soy and 0.1 for beans. 

Also, keeping in mind your comment #5, we have adapted the value of rho to reflect the share of pollinators that are wild pollinators. In the case of forage, even though the level of dependency is set at 1, the fact that we use a value that is less than 1 for rho indicates that, even in the absence of wild pollinators, there will still be production.

3. P28 of 147 Line 271

This approach provides the gross estimate. However, the correct values should be the marginal e.g. without this park nearby farmers may have to hire bees for pollination, they may have lower yields. In either case, you need to properly frame your counter-factual. Or is the counter-factual that these croplands will not produce anything without the park being located within 2km. It should be straight forward to account for high production costs of lower yields in your equation in line 281

Answer: To account for the fact that native pollinators do not represent all of the available pollinators, we adapted the value of rho to reflect the contribution of native pollinators to pollination, based on data from Morse and Calderone (2000). This yields a lower value for the pollination service (10$/ha/yr for rural forests, 14$/ha/yr for urban forests) than what we obtained initially, but it also represents only the pollination service provided by native pollinators.

4. P28 of 147 Line 278

Consider replacing counting be measuring?

Answer: Thank you for making this suggestion; we change the wording accordingly. 

5. P28 of 147 Line 282

The assumption that urban and rural forests have the same pollination contribution is a bit of a weakness. One would have thought that rural forests will be a richer habitat for native bees than urban forests and thus rural forests will have higher populations of bee/insect pollinators?

Could rho (line 281) be used to account for lower native bee pollinators when the park is not there or when it is there but of lower quality compared to its current status.

Also, could rho be used to differentiate between urban and rural forests?

The authors do allude to this fact in line 407-408.

Answer: We went back into our GIS data to redo the analysis to differentiate between rural and urban areas, as we have found a study that calculated wild pollinator populations in gardens in urban (New York City) and suburban areas (Westchester County) and that found a reduction of about half of the population, most likely due to the quality of the nesting environment (Fetridge et al. 2008). Although the population density in the Ottawa-Gatineau region is significantly lower than it is in New York City, and even lower than the density in Westchester county in 2000 (Westchester county: 2091-2692 people/km2; Gatineau-Ottawa: 195.6 people/km2), we decided it was more prudent to reduce the value of rho by half to differentiate between rural and urban environments. 

“In our case, we used the inverse of the values provided by Morse and Calderone [55]; their value reflected the proportion of pollinators that are honey bees. This value, which represent pollinators that are not honey bees, is 0.1 for berries, 0.5 for soybeans and beans, and 0.4 for forage in rural areas. We estimate that this value (ρ) is divided by half in urban areas, considering the quality of the nesting environment [Fetridge et al. 56]. We limited our estimation to crops that have a dependence on pollinators.”

As part of the GIS analysis, we created a 1km buffer around urban agricultural croplands that grow pollinator-dependent crops (beans, berries, pastures / forages, soybeans), and then did the same for crop that were labeled as rural agricultural in our database. There was some overlap between the two buffer layers, as some urban and rural crops were somewhat closely located. As a precaution, we clipped from the rural agricultural buffer all of the area that fell in the urban agricultural buffer. Doing this, however, we ended up with some rural forests in the urban agricultural buffer. This is why the area assigned to urban forests is larger in Table 5 than the total urban forests area. 

6. P29 of 147 Line 294 - 296

Please explain why the Dupras et al values are most suited for your recreation estimates. A brief explanation here will be helpful for the reader.

Answer: In the Dupras et al. study, we used the data on user fees taken from a report by the NCC to estimate the recreational value of ES. The NCC collects user fees only for winter recreational activities, and in the summer, for camping and for parking near beaches and some hiking trails in the Gatineau Park. As a result, this value represents only a portion of the activities undertaken in the Gatineau Park, as there is no general fee to access the park in the summer and a lot of activities can be done for free (hiking, cycling). The collected fees, however, are directly tied to activities carried out in the park. 

Since the other areas of the NCC’s Green Network are all free of access, we cannot compute a value for the recreational ES of the other areas (Urban green network and the Greenbelt). Considering that the activities undertaken in the NCC’s Green Network are similar across land uses, with the exception of croplands, we believe we may use the value obtained for the recreational ES for Forests and Woodlands, Wetlands, Prairies, pastures and grassland, and Freshwater. 

We have reformulated this section to reflect the fact that we will not use the data from the Environics report (27) to estimate the recreational ES, to reflect your comment #1. 

7. P29 of 147 Line 297 to 306

This section only covers the social cost of carbon, how were the biophysical parameters estimated (i.e. tonnes of carbon stored and sequestered)?

Answer: The biophysical parameters were estimated based on other studies. For instance, the carbon storage of forests was estimated using data from the study by Kurz and Apps (86), while the carbon storage of wetlands was estimated using data from Garneau and Van Bellen (98) and the carbon sequestration of wetlands was estimated using data from Lafleur et al. (99). 

These studies are mentioned explicitly in the Results section when discussing specific ecosystems. To make sure this information is clearer for the readers, we added these sentences after lines 297 to 306: 

“To estimate the value of the carbon stored and sequestered by specific features of ecosystems, we relied on estimates from the literature. These specific studies are mentioned in the Results section.”

8. P29 of 147 Line 297 to 305

What was the appraisal period for your discounted cash-flow?

Answer: The net present value of carbon stocks were discounted over 50 years at a 3% discount rate. To make this clearer in the methods section, we added the following sentence:

“To obtain the value of carbon storage annually, we calculated the present value of carbon stocks over 50 years.”

9. P37 of 147 Line 437 to 444

Why was this approach adopted? Both the quality of the environment asset and production value of crops affect the magnitude of the estimated benefit through pollination. Both are important in putting a dollar value on the pollination ecosystem service.

Answer: I am not sure this explanation is very clear in our text. What we did is to divide the production value of crops that are within 1 km of forested areas by the nesting area for pollinators (forests, grassland). This way, we could infer part of the value of production (the part dependent on wild pollinators) to the nesting areas. We modified the text to make it clearer and we added an equation:

“While calculating the value of the pollinating service, we took into account the quality of the nesting environment for wild pollinators, as it is preferable to take into account the quality of the nesting environment, as well as the quality of the agricultural environment [88]. This meant dividing the value of ρ by half for urban crops, according to insights about the quality of urban environments for wild pollinators (Fetridge et al. 2008). The values were obtained by using the sum of the value of pollination for crops grown in rural areas (from Equation 1), divided by the total area in the 1 km buffer around crops that can be considered as nesting habitats for wild pollinators and then multiplied by the share of rural forests in the 1 km buffer (equation 3). We then did the same for urban forests, rural and urban prairies and grasslands.

Pollination ES = (VpollinationRural Total / AreaRural buffer ) * % rural forests in buffer (3)”

10. P38 of 147 Line 464 “calculating” should this be calculated?

Answer: Yes, thank you for noticing this error. 

11. P38 of 147 Line 471

Please name and describe the “three monetary estimates” here.

Answer: We created a table in the Supporting information where all of the values used as part of the benefit transfer are provided in more detail. Although it makes the information less easily accessible to readers, we believe that it lightens the text. In the S1 table, the references are also available (Name of authors, year and number of the reference in the text). To make sure that readers are aware of the existence of this information, we added the following sentence at the end of the subsection “Benefit transfer with adjustment” in the Methods section.

“All of the values used as part of the benefit transfer are described in detail in [S1] table, with the country where the valuation took place, the methodology, the value in $2015 CAD/ha/yr and the ecosystem service(s) studied.”

12. P38 of 147 Line 476

Given that these are benefit transfer values, please explain what they represent within your paper as well as having their source. Even in a benefit transfer paper, it is still very important for your readers to have an idea of what a $318/ha/year for waste treatment means without the need to look at other papers. This comment applies to all your benefit transfer estimates in the paper.

Answer: All of the sources used for the benefit transfer are summarized in Table 3, but they are quite detailed in the Supporting Information, with the name of the authors, the country where the valuation took place, the method used to perform the valuation, as well as the value in ($/ha/year). While putting the detailed information from these studies in the Supporting Information makes it somewhat harder for readers to access, we believe it significantly lightens the text, which is already quite long. 

13. P40 of 147 Line 500

Can you please include an equation showing all variables used for this estimation. It will also be helpful to show the reader your input values for the calculations.

Answer: The table describing the variables is available in the Methods section, more specifically in the Benefit transfer with Meta-Analysis subsection. The equation used for the meta-analysis was not in this section, but we will include it there. 

14. P40 of 147 Line 510-511

While you provide a citation. Please explain the basis for the value here.

Answer: The values described in this paragraph are used to show the difference between the values obtained using the meta-analysis approach and values obtained using other methods. This is why, in this paragraph, we do not explain in a lot of detail how the values were obtained, aside from mentioning their valuation approaches. 

Since this may not be clear, we added a sentence at the beginning of this paragraph: 

“As a way of comparison, we found a number of studies that have used other methods to estimate the value of wetlands.”

15. P40 of 147 Line 527

I am unclear on what approach was used for the Gatineau Park analysis to yield the $3,338/ha/yr.

Also, the $19/ha/year to $10,741/ha/year is a very wide range. Do both numbers represent the willingness to for pay recreation at a park similar to your study area?

Answer: We used the same value obtained for the recreational ES as the one described in the Forests and woodland section. You commented this method in comment 1. We will change this value to the old value that we originally obtained in Dupras et al. [S1]. 

As for the studies mentioned, we recognize that the value are not comparable given the activities undertaken in the wetland areas which included hunting. 

We changed the text in this section to reflect the lack of comparability from other studies and the new value.

“The value of recreational services of $75/ha/year was estimated based on the Gatineau Park analysis. The use of this value is based on the fact that accessible wetlands and their surrounding areas in the NCC Green Network are mainly used for birdwatching, biking, hiking, snowshoeing and cross-country skiing. Shirley’s Bay, in the Ottawa Greenbelt, is the only place where ice fishing is allowed. These are activities that some people pay to access in the Gatineau Park. It is also difficult to estimate the value of the recreational service in wetland areas specifically, due to the limited number of studies and the fact that all of the activities in the Greenbelt, where the largest wetlands are located, are free. In addition, most studies that evaluate the recreational service of wetland include fishing and hunting as part of their activities. In this case, it would be inappropriate to compare our value to these, as these activities are generally not allowed in the study area.”

16. P42 of 147 Line 548

Is this a gross or net value of production?

Answer: Based on the information provided in the study by Dupras et Alam, this value represents the net value of production. 

“For tourism and recreational activities in agricultural lands, we used the tourism benefits associated with rural tourism in the region. We used the income from the agro-tourism of 66 agro-businesses in the region in 2005 (…).[Dupras et Alam 2014:10]” The amount obtained was divided by the agricultural land use in the region and then corrected for inflation. 

17. P42 of 147 Line 549

Please explain the adjustment entails.

Answer: The method used for the adjustment is explained in the methods section. The adjustment is made only to take into account the PPP and the inflation rate. 

“The last step of this economic valuation method was to perform an adjustment to transform values in Canadian dollars of 2015. The first step was to convert the values in Canadian dollars using purchasing power parity conversion tables (OECD stats), a method more precise than simply using exchange rates, because it takes into account the purchasing power of each currency. Then, the values in Canadian dollars were transferred into 2015 dollars using the proper inflation rates.”

18. P43 of 147 Line 555

Should the total in the first row and total value column be 4.592 rather than 4592.0?

Answer: No, this is the sum of the ecosystem services for croplands. 

19. P46 of 147 Line 621

Are the dollar values in a per ha basis

Answer: Yes. We mentioned it in the title (in text), but we will add it in the notation to make this clearer visually, as this is different from the other tables. 

20. P46 of 147 Line 624

What is meant by K$ if thousands please specify fully, K$ often used to refer to constant dollars.

Answer: Yes, K$ represents the value in thousands. We will changed the notation in the table heading to ($ ‘000 CAD 2015). 

21. P48 of 147 Line 624

Should this line start with “66% of the estimated economic value ...”

Answer: This is a good suggestion, thank you. We changed the beginning of the sentence.

---

## [Decision Letter · Decision Letter 2]

22 Dec 2020

The economic value of Canada’s National Capital Green Network

PONE-D-20-10375R2

Dear Dr. L'Ecuyer-Sauvageau,

We’re pleased to inform you that your manuscript has been judged scientifically suitable for publication and will be formally accepted for publication once it meets all outstanding technical requirements.

Kind regards,

Neville Crossman, Ph.D.

Academic Editor

PLOS ONE

Additional Editor Comments (optional):

Reviewers' comments:

Reviewer's Responses to Questions

**Comments to the Author**

1. If the authors have adequately addressed your comments raised in a previous round of review and you feel that this manuscript is now acceptable for publication, you may indicate that here to bypass the “Comments to the Author” section, enter your conflict of interest statement in the “Confidential to Editor” section, and submit your "Accept" recommendation.

Reviewer #1: All comments have been addressed

2. Is the manuscript technically sound, and do the data support the conclusions?

Reviewer #1: Yes

3. Has the statistical analysis been performed appropriately and rigorously? 

Reviewer #1: Yes

4. Have the authors made all data underlying the findings in their manuscript fully available?

Reviewer #1: Yes

5. Is the manuscript presented in an intelligible fashion and written in standard English?

Reviewer #1: Yes

6. Review Comments to the Author

Reviewer #1: Dear authors, Thank you for your reply to comments and for clarifying that you had changed back your method for estimating the benefit of the park based on R2’s comments. I’m happy with the changes and have recommended acceptance for publication. Best of luck.

7. PLOS authors have the option to publish the peer review history of their article (what does this mean?). If published, this will include your full peer review and any attached files.

Reviewer #1: No

---

## [Editor Report · Acceptance letter]

8 Jan 2021

PONE-D-20-10375R2 

The economic value of Canada’s National Capital Green Network 

Dear Dr. L'Ecuyer-Sauvageau:

I'm pleased to inform you that your manuscript has been deemed suitable for publication in PLOS ONE. Congratulations! Your manuscript is now with our production department. 

Kind regards, 

on behalf of

Dr. Neville Crossman 

Academic Editor

PLOS ONE